# Joint evolution of irrigation, the water cycle and water resources under a strong climate change scenario from 1950 to 2100 in the IPSL-CM6

Pedro Felipe Arboleda-Obando[1,2], Agnès Ducharne[1,2], Frédérique Cheruy[2,3], Josefine Ghattas[2]

[1] Laboratoire METIS (UMR 7619, Sorbonne Université, CNRS, EPHE), Paris, France
[2] Institut Pierre Simon Laplace (FR 636, Sorbonne Université, CNRS), Paris, France
[3] Laboratoire de Méteorologie Dynamique (UMR 8539, Sorbonne Université, CNRS), Paris, France

*Corresponding author*: Pedro Felipe Arboleda-Obando (pedro-felipe.arboleda-obendo@univ-grenoble-alpes.fr)

**Abstract.** Irrigation, a key activity for food security, uses local water resources to increase evapotranspiration, creating feedback loops with the atmosphere and water resources. With climate change, it is unclear how irrigation will evolve in the future and how it may influence the evolution of water resources and the water cycle. It is also unclear whether irrigation may be constrained by climate change or water resource shortages. Here, we compare two surface–atmosphere simulations performed with the IPSL-CM6 model from 1950-2100: one with irrigation and one without irrigation. In both simulations, the evolutions of atmospheric radiative forcing, land use, and irrigated areas are taken from CMIP6, which uses a historical dataset for the data before 2014 and the SSP5-RCP8.5 dataset for data after 2014. The two simulations reveal strong global warming and precipitation increases between 1950-2000 and 2050-2100 average values (+5.6 °C and +8.1%, on average, over land with irrigation). Over the same period, our results indicate an increase in irrigation (+76% increase in irrigation in the 2050–2100 compared to the 1950–2000 period), which is in line with a significant expansion of irrigated areas. The influence of irrigation on evapotranspiration in irrigated areas is greater in 2050-2100 than in 1950–2000 (+12% vs. +8%, respectively). Evapotranspiration has also been found to increase in non-irrigated areas near irrigated zones owing to an increase in precipitation under historical and future climate conditions. Water depletion due to irrigation is more intense in the future than in the historical period, although climate change increases water storages and river discharge due to more precipitation in the future. We also identified areas where future environmental conditions can limit irrigation or where irrigation can increase tensions over water use (approximately one-third of irrigated areas, including the Mediterranean basin, California, and Southeast Asia). Our results highlight the importance of considering irrigation in climate projections and future water resources assessments.

## 1 Introduction

Irrigation supports approximately 43% of the world's production on approximately 20% of arable land (Grafton et al., 2017; Siebert and Döll, 2010). As a direct consequence, 70% of human water withdrawal is used for irrigation (between 2657 and

3594 km³ y-¹ in 2000, Pokhrel et al. (2016a)). The key role that irrigation plays today in food production and the corresponding water demand is the result of a significant increase in the irrigated area during the 20th century (a fivefold increase between 1900 and 2005, Siebert et al. (2015a)). This expansion of irrigation may continue in the future, as the replacement of rainfed cropping systems with irrigated systems is one of the measures used to adapt agriculture to climate change (Okada et al., 2018).

In addition to its beneficial effects on food production, irrigation has a direct effect on water and energy balances and surface and subsurface hydrology (Döll et al., 2012; Taylor et al., 2013). Such effects can even drive the evolution of certain variables over time (Al-Yaari et al., 2022; Vicente-Serrano et al., 2019). In addition, the increase in evapotranspiration (ET, due to increased crop transpiration and higher near-surface soil moisture) also induces atmospheric feedback loops such as air temperature cooling (Thiery et al., 2020) and changes in precipitation patterns at different scales that affect the water cycle

(Al-Yaari et al., 2019; Cook et al., 2015; Guimberteau et al., 2012b; Lo and Famiglietti, 2013; de Rosnay et al., 2003; de Vrese et al., 2016a).

With the acceleration of climate change, there are concerns related to the future of irrigation, as well as its effects on water resources under a changing climate. This has led to global projections (Wada et al., 2013a; Khan et al., 2023a) via global hydrology models (GHMs). However, these modeling efforts with GHMs prescribe atmospheric forcing, so the interaction

between irrigation and the atmosphere is not considered.

To consider joint projections for water resources, irrigation, and climate, irrigation must be represented in land surface models (LSMs) within Earth System Models (ESMs), and the LSM must be run in coupled mode with an atmospheric model (McDermid et al., 2023). In recent years, many LSMs have included irrigation modules, such as ORCHIDEE (Yin et al., 2020; Arboleda-Obando et al., 2024a), CLM (Leng et al., 2013; Yao et al., 2022), ISBA-SURFEX (Druel et al., 2022), and MIROC

(Pokhrel et al., 2012, 2015). While some of these LSMs have been used in coupled mode under a historical climate (see Al-Yaari et al., 2022a), coupled simulations under a future climate scenario are less common (Cook et al., 2020). This means that the coupled evolution of irrigation activities, water resources, and climate under a future climate change scenario needs to be further explored.

Here, we present the results of a pair of coupled simulations using the Institut Pierre Simon Laplace Climate Model 6 (IPSL-

CM6), the version used for the Coupled Model Intercomparison Project phase 6 (CMIP6). In Section 2, we present the atmospheric and land surface components of the IPSL-CM6, including the main characteristics of the irrigation scheme; then, we present the modeling setup and analysis methods. In Section 3, we explore how irrigation rates are expected to evolve from 1950-2100. We then evaluate the evolution of the influence of irrigation on variables related to the water cycle and water resources. We also explore the limits to irrigation growth under future hydroclimate conditions to identify areas where tensions

over water use might increase due to irrigation or where irrigation expansion could be possible. Finally, we discuss the limitations of our results and their implications for climate projections and water resources assessments under future climate conditions.

## 2 Methodology

### 2.1 LMDZOR model

We used the LMDZOR model (Cheruy et al., 2020), which involves the coupling of the atmosphere and land components of the IPSL-CM, respectively the LMDZ (Laboratoire de Météorologie Dynamique with Zoom capacity) and ORCHIDEE (Organising Carbon and Hydrology In Dynamic Ecosystems) models. This version uses the LMDZ6A atmospheric model (Hourdin et al., 2020; Sadourny and Laval, 1984) embedded in IPSL-CM6A-LR (Boucher et al., 2020), but we used a medium resolution rather than the standard low resolution (MR and LR, respectively), i.e., 256x256 grid cells, approximately 1.41°x0.71° in size, and 79 vertical levels. Except for the resolution, the configuration remains close to the model used for CMIP6.

The LMDZ model uses a finite difference discretization of the primitive meteorology equation on an Arakawa C grid, favoring the conservation of entropy rather than that of energy (Boucher et al., 2020). It can refine the grid in both longitude and latitude, and the Z in LMDZ indicates its zoom capabilities. Apart from the dynamics, the model couples physical parameterizations of different processes through a generic interface that computes the vertical transfer of those physical processes. The LMDZ6A version includes important improvements in several processes: it computes eddy diffusion by introducing prognostic turbulent kinetic energy (TKE), with particular attention given to the representation of very stable boundary layers over ice sheet plateaus and boreal lands. It also includes a boundary layer convection module, represented by a thermal plume model, with a time implicit scheme and upwind space finite volume scheme for numerical stability and a better representation of stratocumulus clouds. Finally, it also includes a new deep convection module that assumes coupling between shallow convection and deep convection at the cumulus base level, with a statistical estimate of the maximum vertical velocity (Hourdin et al., 2020).

The ORCHIDEE land surface model describes the mass, momentum, and heat fluxes between the surface and the atmosphere (Krinner et al., 2005). The version used here corresponds to ORCHIDEE 2.2, which is similar to the ORCHIDEE 2.0 model used in CMIP6 (Boucher et al., 2020; Cheruy et al., 2020) but includes some minor bug corrections and a global irrigation scheme that was recently developed and evaluated (Arboleda-Obando et al., 2024). The soil column was set to a depth of 2 m, but we used a version with 22 layers instead of 11 layers, so the soil humidity in the root zone defined for the irrigation scheme was finely modeled. ORCHIDEE has been extensively described elsewhere, and we summarize here the main characteristics of the model.

The turbulent fluxes between the surface and atmosphere use the Monin–Obukhov theory and the bulk formulations proposed by Louis et al. (1982). However, the stability functions for the calculations of the surface drag coefficients proposed Louis et al. (1982) are replaced by functions proposed by (King et al., 2001) to better represent stable conditions (Vignon et al., 2017). Additionally, the computation of the surface roughness height has been improved by introducing a dynamic roughness height. Vegetation is represented by 15 plant functional types (PFTs, including bare soil), each with different parameter values to

simulate photosynthesis and carbon allocation; carbon fluxes and plant phenology are controlled by the STOMATE (Saclay Toulouse Orsay Model for the Analysis of Terrestrial Ecosystems; Krinner et al. 2005) module, which computes the evolution of the leaf area index (LAI) . Note that no specific crop phenology module was used, following Arboleda-Obando et al. (2024), owing to a lack of ubiquitous parameters at the global scale. It means that C3 and C4 crops are assumed to have the same phenology as natural grasslands but with higher carboxylation rates. Remark that this simplification may have an impact on the timing and volume of total water withdrawal, by inducing an overestimation of water demand and ultimately water use (if there is available water offer; Arboleda-Obando et al. 2024). This is due to the lack of representation of the crop calendar, including the harvest stage, which keeps the leaf area index values high in the model.

Evapotranspiration follows a classical bulk aerodynamic approach with four subfluxes: snow sublimation, interception loss, bare soil evaporation and transpiration. There are three soil tiles according to vegetation type (bare soil, forest, and crops and grasses) within each grid cell. Surface infiltration is represented as a sharp wetting front based on the Green and Ampt model, whereas vertical soil flow is represented by a 1-D Richards equation (D'Orgeval et al., 2008; Tafasca et al., 2020). The soil is assumed to be homogenous inside the grid cell and is represented by the dominant USDA soil texture according to the map from Zobler (1986). While lateral fluxes between grid cells are neglected, a routing scheme transfers surface runoff and drainage from land to the ocean through a cascade of linear reservoirs (Guimberteau et al., 2012a; Ngo-Duc et al., 2007). Each grid cell is split into transfer units (also called subbasins) with the river reservoir and two local reservoirs: overland and groundwater. Subbasins are defined according to a flow direction map from Vörösmarty et al. (2000) and are enhanced over the polar region by Oki et al. (1999). Note that owing to the coarse resolution, it is possible to have more than one transfer unit inside every grid cell.

While the water balance is independent for each soil tile, the energy balance is the same for the whole grid cell. The surface energy and water budget computation time are the same as that of the atmospheric model, which is 15 minutes. The routing scheme also uses a 15-minute time step (the standard is one day) to finely follow changes in reservoirs due to water withdrawal. In contrast, the carbon and plant phenology computed by STOMATE uses a daily time step.

## 2.2 Irrigation scheme

The irrigation scheme used here was tested and evaluated in Arboleda-Obando et al. (2024a) at the global scale. This work included a sensitivity analysis to understand the effect of each parameter on irrigation volume and evaporation increase, leading to choosing a set of globally homogenous parameters enabling a good match between simulated and reported values of global irrigation withdrawal. Here, we briefly describe its main characteristics. First, the root zone depth is set according to a user-defined parameter. In our case, we set this depth to 0.65 m (11 layers). This depth comprises approximately 90% of the crop root system as represented in ORCHIDEE. Then, at each time step, the scheme calculates a soil moisture deficit. This deficit is the difference between the actual soil moisture and a user-defined target in all the root zone layers. We set this target to 0.9 times the soil moisture at field capacity. Third, we calculate the irrigation requirement using the prescribed fraction of irrigated grid cells and limit the maximum irrigation per hour to 3 mm/h.

In the fourth step, the module estimates the available water in the natural reservoirs (overland, groundwater, and river reservoirs), with two constraints: a factor to prevent total depletion of the natural reservoir and mimic an environmental flow (set to 0.9 for all three reservoirs) and the facility for accessing the natural reservoir as represented by a prescribed map of irrigated areas that are equipped for surface or groundwater use (Siebert et al., 2010). The facility of water access helps to prioritize one reservoir over others; for instance, in a gridcell with irrigation demand and groundwater availability but no groundwater access infrastructure, all the water supply will come from the surface reservoirs (overland and stream). More details can be found in Arboleda-Obando et al. (2024). Apart from the local reservoirs, the scheme allows river water adduction from local transfer units (also called subgrid basins in Ngo-Duc et al. 2007) within the grid cell, limiting the available water with a factor set here to 0.05 to prevent river depletion. Owing to the coarse scale, water adduction from neighboring grid cells is deactivated.

In the final step, the scheme estimates the maximum water requirement and water supply (locally available water and adduction) and withdraws the volume from the natural reservoirs. The withdrawn water is added at the soil surface for infiltration, thus resembling a flood or drip irrigation technique. This implies that not all of the water volume is involved in transpiration or bare soil evaporation, as the model decides if part of this volume becomes surface runoff or groundwater recharge. The scheme does not represent paddy rice as a different irrigation technique and uses global homogenous parameters taken from Arboleda-Obando et al. (2024a).

We note here some shortcomings identified in the irrigation scheme used here (Arboleda-Obando et al., 2024). The scheme represents a single irrigation technique (the flood technique), and uses a set of simplified rules to trigger irrigation and allocate available water. Besides, the scheme uses a joint representation of rainfed and irrigated crops within the same tile, and the scheme doesn't represent conveyance losses. To restrain in part the effect of these shortcomings on estimated irrigation volumes, parameter values were tuned by fitting the simulation to reported irrigation datasets. But we must also note that this parameter tuning is overly simplistic, as it uses globally uniform parameters. Despite these limitations the irrigation scheme produces acceptable estimates of yearly estimation withdrawals at global scale, but tends to underestimate irrigation withdrawals in China, India and the USA, corresponding to the irrigation hotspots (Arboleda-Obando et al., 2024).

**2.3 Radiative forcing and socioeconomic scenario**

We ran two simulations, one without the irrigation scheme (NoIrr simulation) and a second with the irrigation scheme activated (Irr simulation), for the period 1950 to 2100. Each simulation runs 50 years of spin-up with a prescribed sea surface temperature/sea ice content (SST/SIC) from the AMIP dataset and historical radiative forcing.

The radiative forcing is prescribed using historical (1950-2014) and SSP5-RCP8.5 (2015-2100) datasets from ScenarioMIP (Tebaldi et al., 2021). The use of scenario SSP5-RCP8.5 could be seen as the upper boundary of potential climate change impacts and results in a strong global warming and important changes in precipitation. The use of this scenario allows for analysis of the interaction between irrigation and climate in a context of strong climate change signals and significant changes in irrigated land area. For oceanic conditions, the simulations were run with a prescribed bias and variance-corrected SST/SIC

dataset (Beaumet et al., 2019) constructed on the basis of the SST/SIC simulated by the fully coupled IPSL-CM6 and observed AMIP. This fully coupled simulation uses historical and SSP5-RCP8.5 data to prescribe radiative forcing as well. The unbiased reference is taken from the AMIP dataset.

The land use is prescribed for each year using the Land Use Harmonization 2 (LUHv2) (Hurtt et al., 2020) dataset from the
Coupled Model Intercomparison Project 6 (CMIP6); we used the historical and SSP5-8.5 scenarios for 1950-2014 and 2015-2100, respectively. Changes in land use include changes in cropland area (Figs. S1 and S2 in the Supplementary Material). Each year of irrigated area per grid cell is also prescribed with LUHv2, using the same setup setup (historical and the SSP5-8.5).

The average spatial distribution of irrigated fractions for the 2050—2100 period (Figure 1-a) includes the current hot spots
(India, China, Southeast Asia, and the USA) in addition to new hot spots in Africa (Southeast Africa) and South America (Rio de la Plata area). Changes in the irrigated fraction between the future (2100-2050) and the historical (2000-1950) periods show a strong increase in the irrigated fraction in the new hotspots (Africa and South America) and in northern India and Southeast Asia (Figure 1-b), but Asia remains the main hotspot of irrigated areas (Figure 2-a).

Note that some areas do not depict major changes or that the irrigated area even decreases (Iraq and some areas in the
Mississippi River Basin). Additionally, scenario SSP5-8.5 assumes that the ratio of irrigated area to cropland area increases (Fig. S2), despite an increase in cropland area, indicating a greater role for irrigation in agriculture and food production. To prescribe the factors of facility of access, we used the Siebert et al. (2010a) map of the fraction of irrigated area equipped with surface water, following Arboleda-Obando et al. (2024a). This map is fixed and representative of conditions around the year 2000. This means that there is no adaptation process related to changes in infrastructure, such as a shift from surface water use
to groundwater use.

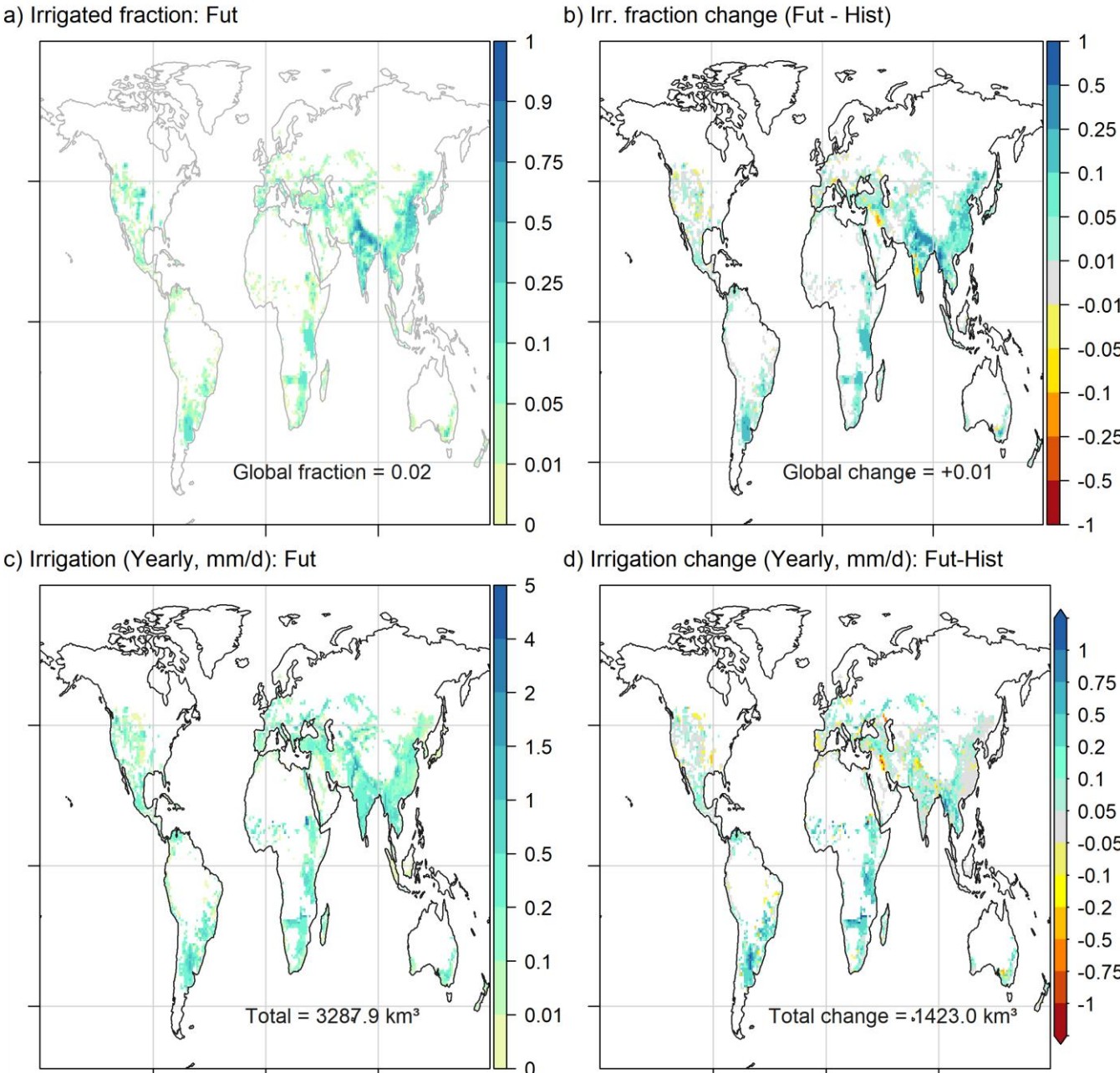

**Figure 1: (a) Map of the average irrigated fraction by grid cell for the future period (2050-2100). (b) Map of the change in the average irrigated fraction by grid cell between the future (2050-2100) and historical (1950-2000) periods. The white areas correspond to grid cells with no irrigated fraction. The irrigated fractions are prescribed by the LUHv2 dataset and interpolated to the model resolution. (c) Map of the yearly average irrigation for the future period (2050-2100). (d) Map of the change in the yearly average irrigation amount between the future (2050-2100) and historical (1950-2000) periods. The white areas correspond to grid cells with no irrigation.**

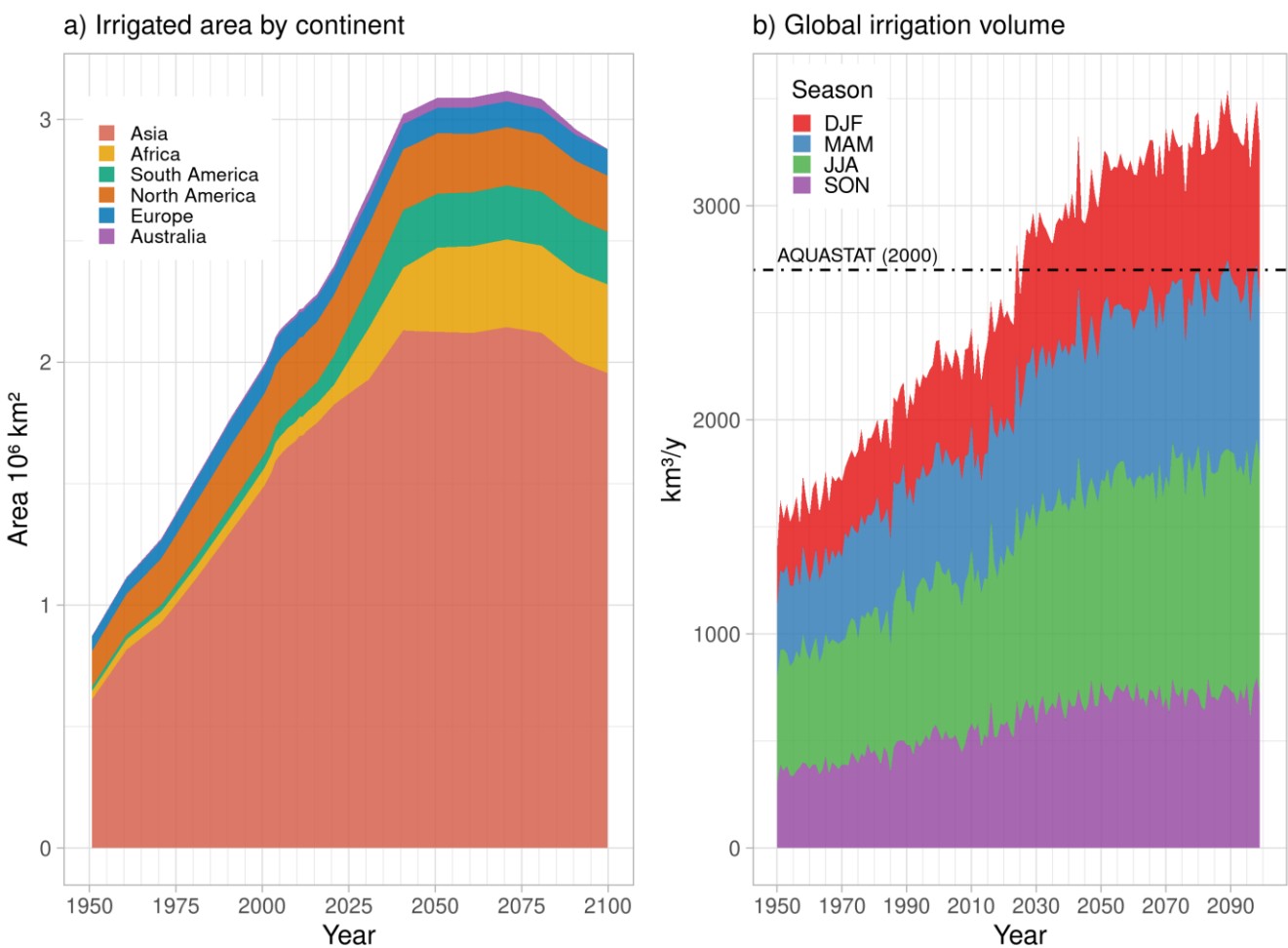

Figure 2: (a) Total irrigated area and by continent prescribed by LUHv2. (b) Total and seasonal irrigation volume at the global scale simulated by IPSL-CM6 in the Irr simulation. The black line represents the value reported by AQUASTAT around the year 2000 (Frenken and Gillet, 2012).

### 2.4 Analysis tools

We focused on average changes over land, in irrigated areas and in non-irrigated areas, for irrigation as well as important land and atmospheric variables related to the water cycle. Land refers here to all continental areas except Greenland and Antarctica, which are not represented by the ORCHIDEE LSM. We considered that a grid cell belongs to the "irrigated areas" if the average irrigated fraction from 1950-2100 was different from zero (see shaded grid cells in Figure 1a and b) or to the "non-irrigated areas" if the average irrigated fraction was equal to zero for the same period (see white grid cells in Figure 1a and b). We analyze here three main changes on the basis of the differences between the two simulations and between two different periods:

1.  The climate change impact is defined by the difference, for a given simulation (Irr or NoIrr), between the future and historical periods (Fut and Hist, set here as 2050-2100 and 1950-2000, respectively). In the following, the impact of climate change is assessed mainly on the basis of the Irr simulation, i.e., Irr(Fut) - Irr(Hist).

2.  The influence of irrigation during a given period is identified as the difference between the Irr and NoIrr simulations for a given period. In the following, the influence of irrigation, such as Irr(Fut) - NoIrr(Fut), is assessed in the future period because the influence of irrigation is stronger in the future period (Section 3.2).

3.  Since the evolutions of irrigation and climate are coupled in the Irr simulation, we finally introduce a coupling metric, called modulation. This modulation can be described as the effect of irrigation on the alterations produced by climate change, i.e., [Irr(Fut) - Irr(Hist)] - [NoIrr(Fut) - NoIrr(Hist)], but this modulation is equivalent to the effect of climate change on the alterations produced by irrigation, i.e., [Irr(Fut) - NoIrr(Fut)] - [Irr(Hist) - NoIrr(Hist)]. This modulation term is similar to the one introduced in Arboleda Obando et al. (2022) to characterize the coupled changes of climate change and hillslope flow on the basis of trends. While in this study there is no representation of hillslope flows, the concept of modulation is still useful to understand the dynamic interaction between a process in the continental component of the climate model, and a changing climate.

The statistical significance of the average differences between the two periods (for climate change impacts and modulation) and between the simulations (for irrigation influence) was evaluated with a Student's t test at the 5% significance level.

## 3 Results

### 3.1 Evolution of irrigation under a changing climate

In the SSP5-RCP8.5 framework, irrigation continues to increase throughout the simulation period (+76% in the future, for years 2050-2100; compared with the historical period, 1950-2000). At the seasonal scale, global irrigation increases for all four seasons, and the relative weight of each season is not markedly changed throughout the simulation period, although the weight of DJF increases slightly to the expense of MAM; JJA remains the main irrigation season (around 1000 km$^3$, Figure 2-b). This is consistent with Asia remaining the main irrigation hotspot despite irrigation expansion in Africa and South America (which are located in the southern hemisphere). For the future period, irrigation uses a volume of 3280 km³ (compared with 2700 km³ reported by AQUASTAT around the year 2000), and the most intensively irrigated areas correspond to the Indus and Ganges River Basins, the Nile River Basin, and Southeast Asia River Basins (Mekong, Yellow River, and Yangtze; Figure 1-c).

The map of the differences between the future and historical irrigation rates (Figure 1-d) reveals a more contrasting distribution. We observe three main classes: first, areas with an increase in irrigation (southern and central Africa, southern South America, Southeast Asia); second, areas with no major change in the irrigation rate (e.g., China, southern India, and central USA); and third, areas with decreasing irrigation (e.g., the Iberian Peninsula, Iraq, and the Mississippi River Basin). The relationship between a decreasing irrigated area and a decreasing irrigation rate is clear in some areas (see the Mississippi River Basin and

Iraq); however, other areas are less irrigated despite the expansion of irrigated areas (see areas in the upper Indus river basin, in the lower Ganges river basin and in the Iberian Peninsula). Additionally, an important increase in the irrigated area does not necessarily translate to an important increase in irrigation, as seen in China. This means that climate factors could contribute to explain the evolution of irrigation. But it should be noted that model choices, such as the use of global parameters, a simplified representation of irrigation and the lack of crop phenology, can influence the magnitude or even the sign of water demand and thus the evolution of irrigation.

## 3.2 Average changes and modulation

Table 1 shows the average values from both simulations of 10 key hydroclimate variables, the influence of irrigation and climate change impact on those variables, and the modulation for land, irrigated areas and non-irrigated areas. In section 3.4 Irrigation influence we assess the spatial distribution of the influence of irrigation for P, ET and water resources. Climate change accelerates the water cycle, warms the air, and increases net radiation. We observe that the influence of irrigation (i.e. the difference between the NoIrr and the Irr simulations) increases the average land values of ET, precipitation (P), runoff (R), and LAI (+4%, +1%, +2% and +3%, respectively),  while it depletes water storage in irrigated areas, i.e., groundwater storage (GWS, -7%) and stream storage (Stream S, -8%), but increases water storage in non-irrigated zones (+2% for GWS). The influence of irrigation on total water storage (TWS) on average land values is positive (+0.7%), which is partially due to an increase in soil moisture (SM, +1% over land).

In the case of water-related variables and the LAI, the effect is stronger in irrigated areas, but there is a positive effect in non-irrigated areas owing to a positive atmospheric response that increases P by approximately 1% under historical and future climates. On the other hand, the influence of irrigation is positive for net radiation and negative for air temperature at 2 m (Tas) and is confined to irrigated areas. Changes in radiation variables are explained by evaporative cooling and a decrease in longwave radiation emission, which increases the net radiation.

Modulation is important in irrigated areas for the ET, R, and water storage variables but rather weak in non-irrigated areas. For example, irrigation increases ET by 8% under historical conditions and by 12% under future climate conditions in irrigated areas, which means that irrigation accelerates the increase in ET induced by climate change in the Irr simulation. Net radiation and air temperature evolution are also affected by irrigation in irrigated areas, but there is no major change in the evolution in non-irrigated zones when comparing Irr and NoIrr. On the other hand, the influence of irrigation on the precipitation evolution is negligible in both irrigated and non-irrigated areas.

## 3.3 Climate change impacts

The spatial distributions of the impacts of climate change on precipitation and air temperature are shown in Figure 3-a and b (Fig. S3 for the spatial distributions, including those of the oceans). In irrigated areas, precipitation may either increase due to climate change (e.g., China and southern India, with local extremes of more than +1 mm/day) or decrease (Mediterranean area,

with local extremes in the Iberian Peninsula below -1 mm/day), whereas warming occurs in all areas. These changes in climate can contribute to changes in irrigation: positive changes in precipitation can increase available water and water resources while decreasing the soil moisture deficit and water demand. Negative changes in precipitation increase water demand, which could increase irrigation if water resources are available. Warming tends to increase water demand, but it should be noted that warming tends to be greater in northern latitudes (warming above 7 °C) than in tropical and southern areas(warming above 3

270 °C)  as a result of the land warming pattern, visible in non-irrigated areas (Figure 3-b).

The impacts of climate change present similar spatial distributions to those of other hydrologic variables, with some differences (Figure 4). ET increases in the future compared with the historical period in the Irr simulation (Figure 4-a), with major exceptions in a few tropical regions (central Africa and parts of the Amazon River basin) and at higher latitudes (the Iberian Peninsula and southern Australia) owing to changes in net radiation and precipitation, respectively. The impact of climate

change on runoff (R, Figure 4-b) follows a spatial pattern similar to that of P changes, with some exceptions in North America. Finally, climate change impacts on water storage, i.e., GWS and Stream S (Figure 4-c and d), follow the spatial patterns of P. Note that there are exceptions for GWS in irrigated areas such as the central U.S. and South Asia, which depict negative changes even if precipitation increases. Additionally, the strongest changes in stream reservoirs are found in grid cells containing the largest rivers, since changes in the stream water budget of any grid cell propagates along the river network, and

accumulate in the grid cells with large upstream areas (e.g. Amazon river, Nile river, Congo river and Indus river). The next step is to assess in more detail the influence of irrigation on key hydroclimate variables.

Table 1: Yearly average values over land, irrigated areas and non-irrigated areas for 10 variables related to hydroclimate surface conditions. The considered values by variable are the average in the irr simulation for a chosen period (Hist, 1950-2000 and Fut, 2050-2100), the influence of irrigation by period, the effect of climate change in the Irr simulation, and the modulation. The values in bold marked with an * correspond to a p value under 0.05 according to Student's t test.

| Variable (yearly) | Simulation | All land | | | Irrigated regions | | | Non-irrigated regions | | |
|---|---|---|---|---|---|---|---|---|---|---|
| | | 1950-2000 | 2050-2100 | Fut-Hist | 1950-2000 | 2050-2100 | Fut-Hist | 1950-2000 | 2050-2100 | Fut-Hist |
| ET, mm/d | Irr | 1.66 | 1.72 | **0.06 (3%)\*** | 1.9 | 2.1 | **0.2 (7.9%)\*** | 1.57 | 1.6 | **0.03 (2%)\*** |
| | NoIrr | 1.62 | 1.66 | **0.04 ( 2%)\*** | 1.75 | 1.85 | **0.1 (3.1%)\*** | 1.56 | 1.59 | **0.03(2%)\*** |
| | Irr-NoIrr | **0.04 (2%)\*** | **0.06 (4%)\*** | **0.02 (57%)\*** | **0.15 (8%)\*** | **0.25 (12%)\*** | **0.1 (65%)\*** | **0.008 (0.5%)\*** | **0.009 (0.6%)\*** | 0.001 (12.4%) |
| P, mm/d | Irr | 3.04 | 3.29 | **0.25 (8%)\*** | 3.16 | 3.43 | **0.27 (9%)\*** | 3 | 3.25 | **0.25 (8%)\*** |
| | NoIrr | 3.04 | 3.29 | **0.25 ( 8%)\*** | 3.16 | 3.43 | **0.27 ( 9%)\*** | 3 | 3.25 | **0.25 ( 8%)\*** |
| | Irr-NoIrr | **0.03 (0.9%)\*** | **0.04 (1%)\*** | 0.01 (30%) | **0.04 (1%)\*** | **0.05 (1.4%)\*** | 0.01 (38%) | **0.03 (0.9%)\*** | **0.04 (1%)\*** | 0.01 (26%) |
| GWS, mm | Irr | 34.7 | 41.5 | **6.8 (20%)\*** | 35 | 37 | **2 (6%)\*** | 34.5 | 42.8 | **8.3 (24%)\*** |
| | NoIrr | 35 | 42 | **7.0 (20%)\*** | 37.5 | 42.5 | **5 ( 12%)\*** | 34.1 | 41.9 | **7.8 ( 23%)\*** |
| | Irr-NoIrr | -0.3 (-0.7%) | **-0.5 (-1%)\*** | -0.2 (85%) | **-2.5 (-7%)\*** | **-5.5 (-14%)\*** | **-3 (102%)\*** | **0.4 (1%)\*** | **0.9 (2%)\*** | 0.5 (115%) |
| Stream S, mm | Irr | 7 | 8.7 | **1.7 (24%)\*** | 6.8 | 8.3 | **1.5 (22%)\*** | 7.14 | 8.88 | **1.74 (24%)\*** |
| | NoIrr | 7.1 | 8.9 | **1.8 ( 26%)\*** | 7.4 | 9.3 | **1.9 (26%)\*** | 7.07 | 8.88 | **1.8 ( 25%)\*** |

| | | | | | | | | | |
|---|---|---|---|---|---|---|---|---|---|
| | Irr-NoIrr | **-0.08 (-1%)*** | **-0.2 (-3%)*** | **-0.12 (186%)*** | **-0.6 (-8%)*** | **-1 (-12%)*** | **-0.4 (81%)*** | 0.062 (0.9%) | 0.002 (0.02%) | -0.06 (-97%) |
| R, mm/d | Irr | 1.42 | 1.64 | **0.22 (16%)*** | 1.37 | 1.6 | **0.23 (17%)*** | 1.43 | 1.65 | **0.22 (16%)*** |
| | NoIrr | 1.4 | 1.61 | **0.21 (15%)*** | 1.33 | 1.53 | **0.2 (15%)*** | 1.41 | 1.63 | **0.22 (15%)*** |
| | Irr-NoIrr | **0.02 (2%)*** | **0.03 (2%)*** | **0.01 (54%)*** | **0.04 (3%)*** | **0.07 (4%)*** | **0.03 (87%)*** | **0.02 (1%)*** | **0.03 (2%)*** | 0.01 (35%) |
| LAI, m²/m² | Irr | 1.47 | 1.9 | **0.43 (29%)*** | 1.5 | 2 | **0.5 (34%)*** | 1.45 | 1.85 | **0.4 (27%)*** |
| | NoIrr | 1.44 | 1.85 | **0.41 (28%)*** | 1.4 | 1.8 | **0.4 (31%)*** | 1.45 | 1.84 | **0.39 (27%)*** |
| | Irr-NoIrr | **0.03 (2%)*** | **0.05 (3%)*** | **0.02 (94%)*** | **0.1 (6%)*** | **0.2 (9%)*** | **0.1 (82%)*** | **0.004 (0.3%)*** | **0.012 (0.7%)*** | **0.008 (169%)*** |
| SM, mm | Irr | 493.5 | 498.5 | **5 (1%)*** | 498 | 505 | **7 (1%)*** | 492 | 496 | **4 (0.8%)*** |
| | NoIrr | 490 | 493 | **3 ( 0.7%)*** | 490 | 490 | 0.01 ( 0.003%) | 489.9 | 494 | **4.1 ( 0.9%)*** |
| | Irr-NoIrr | **3.5 (0.8%)*** | **5.5 (1%)*** | **2 (40%)*** | **8 (1.7%)*** | **15 (3%)*** | **7 (85%)*** | **2.1 (0.5%)*** | **2 (0.5%)*** | **-0.1 (-6%)*** |
| TWS, mm | Irr | 690 | 706 | **16 (2%)*** | 692 | 713 | **21 (3.0%)*** | 689 | 704 | **15 (2%)*** |
| | NoIrr | 686 | 701 | **15 ( 2%)*** | 685 | 703 | **18 ( 2.7%)*** | 686.2 | 700.8 | **14.6 ( 2.1%)*** |
| | Irr-NoIrr | **4 (0.6%)*** | **5 (0.7%)*** | 1 (22.6%) | **7 (1%)*** | **10 (1%)*** | **3 (34%)*** | **2.8 (0.4%)*** | **3.2 (0.5%)*** | 0.4 (14%) |
| Net rad., W/m² | Irr | 90 | 95 | **5 (5%)*** | 100 | 105 | **5 (5%)*** | 86.9 | 91.8 | **4.9 (6%)*** |

| | | | | | | | | | |
|---|---|---|---|---|---|---|---|---|---|
| | NoIrr | 89.6 | 94.4 | **4.8 ( 5%)*** | 98.4 | 102.6 | **4.2 ( 4%)*** | 86.8 | 91.8 | **4.9 ( 6%)*** |
| | Irr-NoIrr | **0.4 (0.5%)*** | **0.6 (0.6%)*** | **0.2 (38%)*** | **1.6 (2%)*** | **2.4 (2%)*** | **0.8 (54%)*** | 0.08 (0.09%) | 0.04 (0.04%) | -0.04 (-49%) |
| Tas, °C | Irr | 13.5 | 19.1 | **5.6 *** | 16.5 | 21.5 | **5 *** | 12.5 | 18.3 | **5.8 *** |
| | NoIrr | 13.5 | 19.2 | **5.7 *** | 16.7 | 21.9 | **5.2 *** | 12.5 | 18.4 | **5.9 *** |
| | Irr-NoIrr | **-0.09 *** | **-0.14 *** | -0.05 (57%) | **-0.2 *** | **-0.4 *** | **-0.2 (66%)*** | -0.05 | **-0.07 *** | -0.02 (44%) |

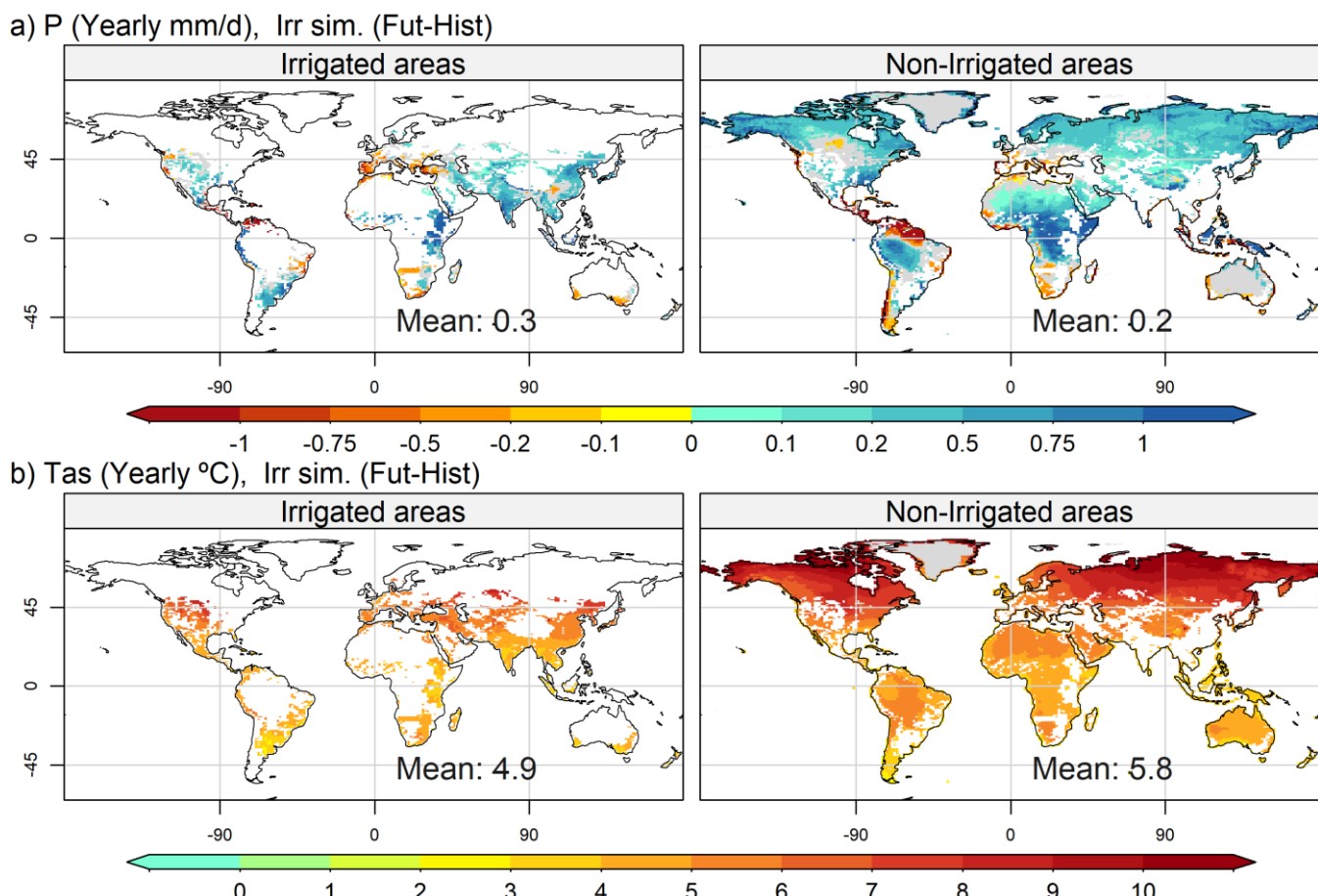

**Figure 3: Map of the spatial distribution of yearly changes between future and historical periods for irrigated areas (left column) and non-irrigated areas (right column), for precipitation (a) and air temperature at 2 meters (b). The areas in gray correspond to a p value less than 0.05 according to Student's t test.**

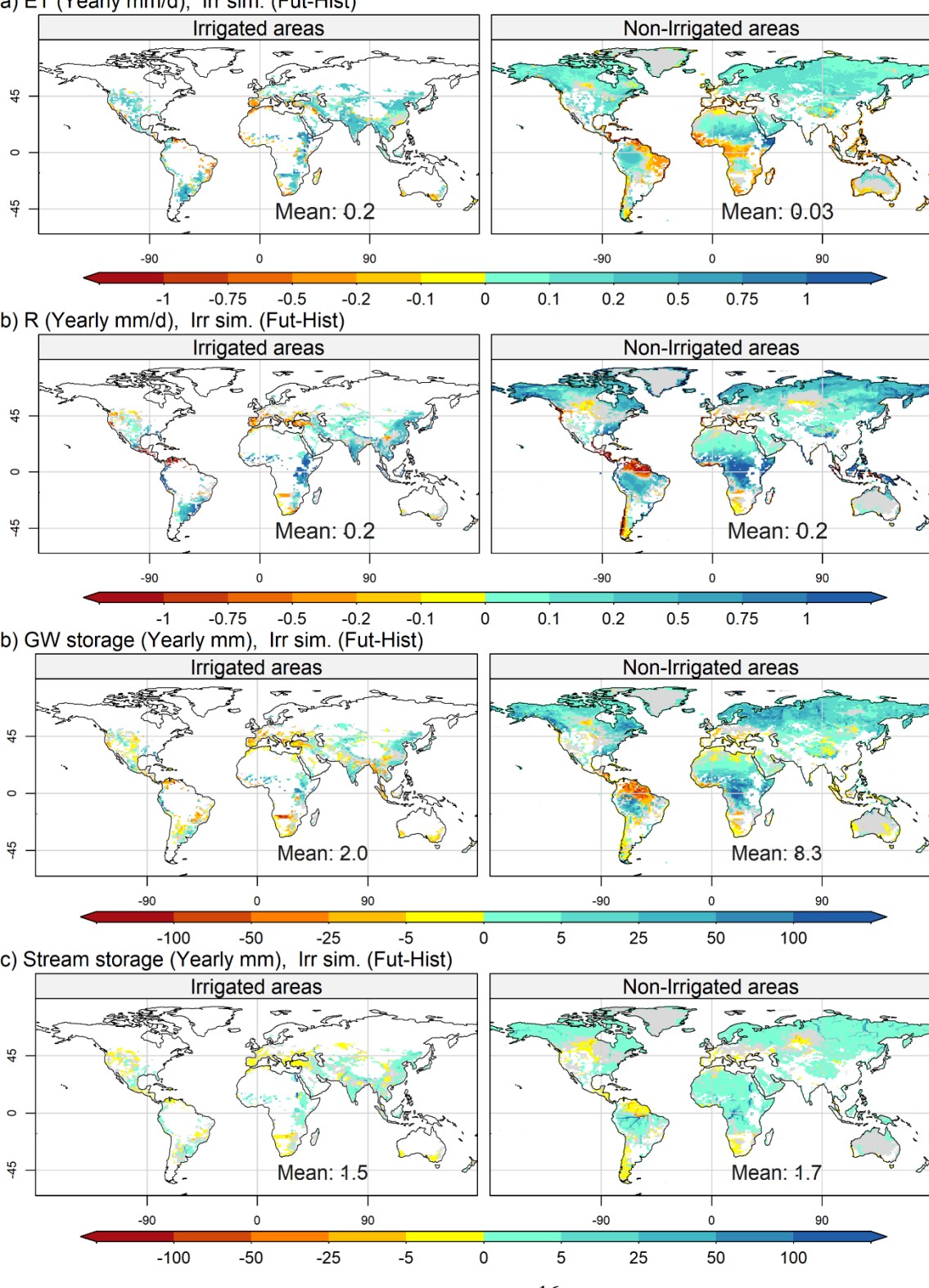

**Figure 4: Map of the spatial distribution of yearly changes between future and historical periods for irrigated areas (left column) and non-irrigated areas (right column), for evapotranspiration (a) total runoff (b), groundwater reservoirs (c) and stream reservoirs (d). The areas in gray correspond to a p value under 0.05 according to a Student's t-test.**

## 3.4 Irrigation influence

The evolution of the ET and P yearly average rates over land, for irrigated areas and non-irrigated areas, as defined in section 2.4 Analysis tools, is shown in Figure 5. For ET, the NoIrr simulation shows a decreasing trend in irrigated areas (from 1.81 mm/d during 1950-1975, to 1.79 mm/d during 1975-2000) during the 1950-2025 period that is not present in the Irr simulation (Figure 5a, second row). Additionally, the changes in ET observed over land (Figure 5a, first row) are driven by changes in irrigated areas (+0.15 mm/d in historical, +0.25 mm/d in future), as the ET values in non-irrigated areas are similar for both simulations (Figure 5a, third row). Finally, we observe that the increase in ET in irrigated areas after 2025 is faster in the Irr simulation than in the NoIrr simulation, even though irrigation expansion stops by 2040 (Figure 2-a). In the case of P, irrigation activities increase the yearly average values over land around 0.04 mm/d (Figure 5b, first row) and in irrigated areas (Figure 5b, second row), but there is no major influence on the evolution over time (both NoIrr and Irr simulations show a similar positive trend over the period). This means that the positive trend in precipitation over land (Figure 5b, first row) is driven by other forcings, i.e., climate change and land use and land cover change.

Figure 6 shows the spatial distribution of the influence of irrigation (Irr-NoIrr) in the future for ET and P. In the future period, irrigation always increases ET in irrigated areas (with local extremes in the Indus river basin above +1 mm/d) , and in many non-irrigated areas nearby (Figure 6-a, especially in central Asia and the African Sahelian band, with values below +0.1 mm/d). We also note non-irrigated areas where ET decreases in the future in the Irr simulation (Russian tundra, some areas in central Africa), involving negative feedback loops between P and ET changes via net short-wave radiation (not shown): in these areas, P increases, either statistically significantly or not, which increases cloudiness and reduces downwelling short-wave radiation, thus reducing ET, as already described in central Africa by Wang et al. (2018); in Northern Russia, the P increase can also lead to increased snowfall and surface albedo, reducing available energy for ET.

Like for ET, irrigation mostly increases P, over both irrigated and non-irrigated areas, but on smaller surfaces (Figure 6-b, with values below +0.5 mm/d; the same figure including the oceans is shown in Fig. S3). The joint increase of ET and P in non-irrigated areas around irrigated areas reveals a remote impact of irrigation linked to atmospheric transport of moisture from irrigated to surrounding areas, which supports higher P and therefore higher ET in non-irrigated areas, such as the Sahelian band and Central Asia. Changes in P farther away from irrigated areas are rare and may result from various atmospheric processes in a generally more humid atmosphere.

The modulation is mostly positive for ET in irrigated zones (indicating that the influence of irrigation on ET is more important in the future; Fig. S4-a) and is similar to the evolution observed for LAI (Fig. S5 and S6). For P, the modulation is weak (Fig. S4-b for P). The influence of irrigation on other variables is consistent with these changes (Fig. S5 and S6 for R; Fig. S7 and S8 for TWS and SM; Fig. S9 and S10 for net radiation and Tas).

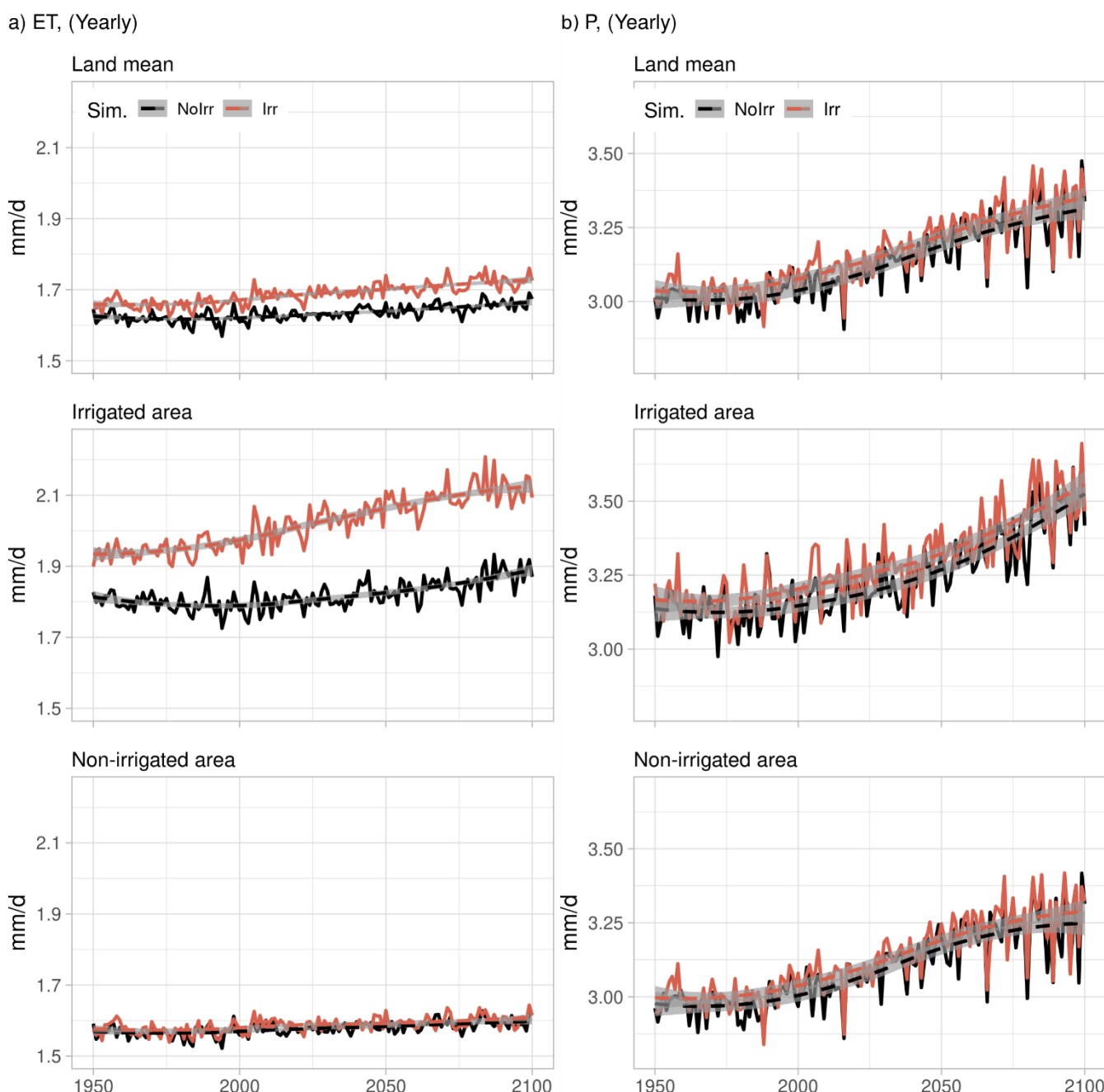

**Figure 5: Time series of yearly evapotranspiration (a, left column) and precipitation (b, right column). The first row corresponds to averages over land, the second row corresponds to averages over irrigated areas, and the third row corresponds to averages in non-irrigated areas. The dashed lines correspond to a fitted polynomial curve via local fitting.**


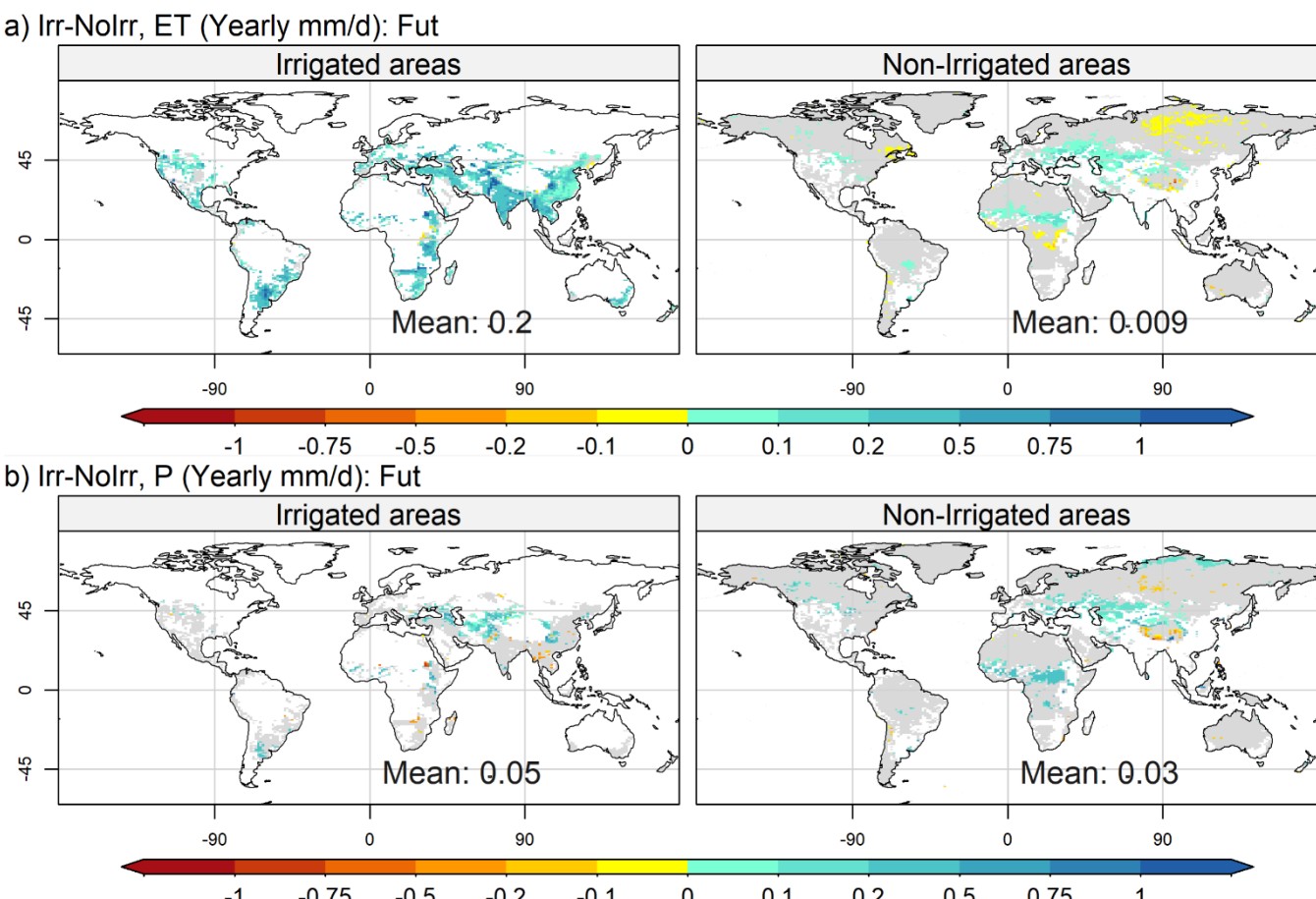

**Figure 6: Map of the spatial distribution of yearly changes between the Irr and NoIrr simulations under future climate conditions for irrigated areas (left column) and non-irrigated areas (right column), for evapotranspiration (a) and precipitation (b). The areas in gray correspond to a p value less than 0.05 according to Student's t test.**

Time series of water storage in groundwater (which represents shallow aquifers, Figure 7a) and stream (which represents large rivers, Figure 7b) reservoirs show important differences between NoIrr and Irr simulations. These differences are explained by complex interactions between irrigation activities, climate conditions, and water resources (see differences over land, Figure 7 first row; irrigated areas, Figure 7 second row; and non-irrigated areas, third row) that we pass to show. The impact of climate change induces a positive trend in water storage Figure 7-a and b, first row), whereas irrigation decreases the average GWS and Stream S in irrigated areas in around -14 and and -12% in the future, respectively (Figure 7-a and b, second row) and slightly increases the GWS and Stream S in non-irrigated areas in around +2 and +0.02% in the future (Figure 7-a and b, third row). The negative effects in irrigated areas are explained by direct water use to sustain irrigation activities, whereas the increase in water resources in non-irrigated areas in the Irr simulation is explained by the fact that irrigation increases precipitation remotely, in particular around irrigated zones (Figure 6b, right panel). The modulation is also negative in irrigated

areas (i.e., water resource exploitation increases) but weak in non-irrigated areas. Note that the effect of irrigation on water storage reservoirs seems to counteract the increase induced by climate change in irrigated areas before 2040, and after 2040 the increase of water storage (groundwater and stream reservoirs) is slower in the Irr simulation than in the NoIrr simulation.

Figure 8 depicts the spatial distribution of the influence of irrigation in the future for GWS and Stream S. The effects are mostly negative for both variables in irrigated areas, but in the case of Stream S, depletion is more important in the grid cells containing large rivers (with local extreme values under -50 mm in the Indus and Rio de la Plata rivers) because the influence of irrigation propagates through the river system. Additionally, the modulation is mostly negative for both reservoirs (with some local exceptions in the GW reservoir), indicating that the water use intensity increases during the simulated period (Fig.

S4-c and d).

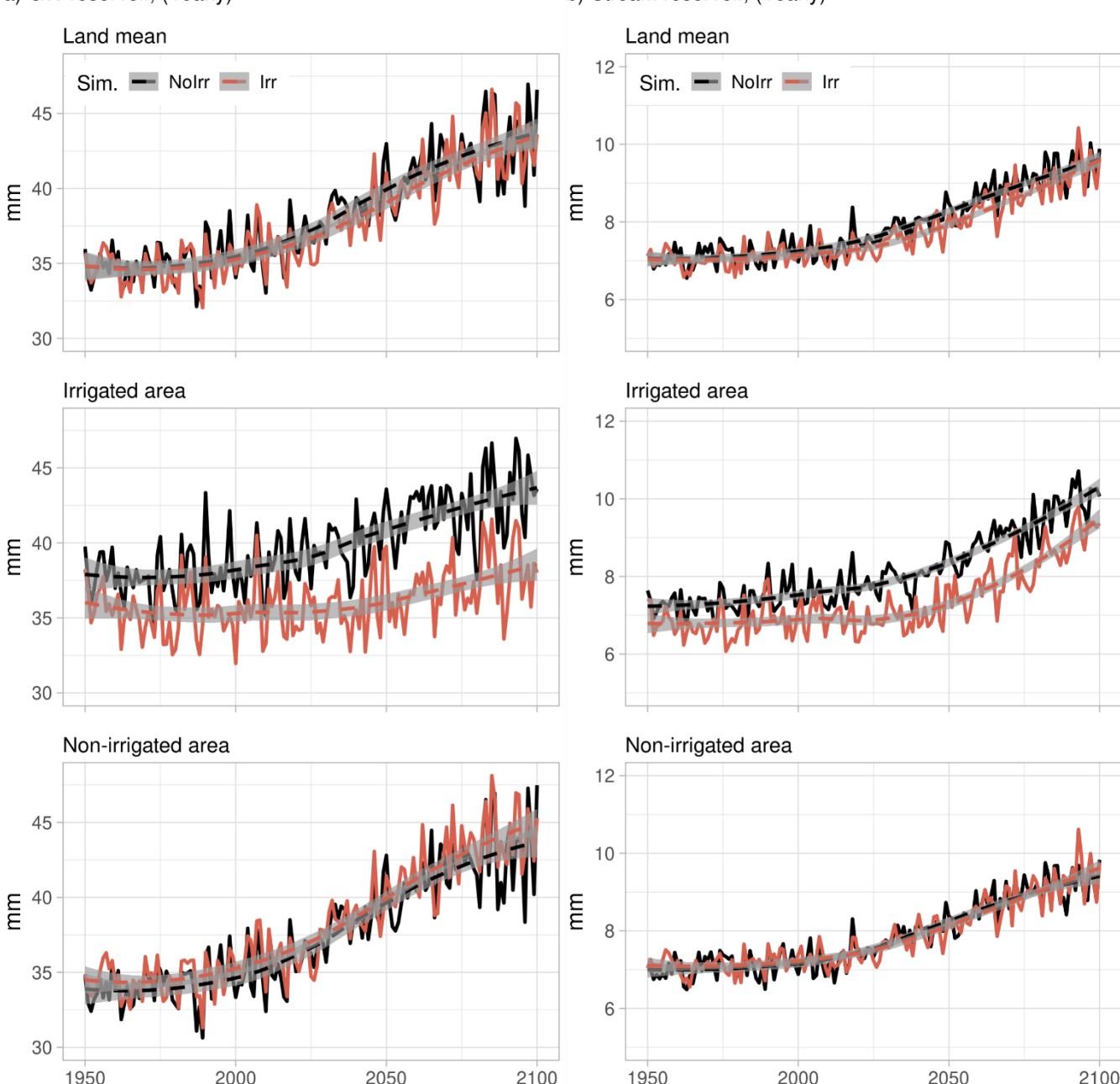

**Figure 7: Time series of yearly groundwater storage (a, left column) and stream storage (b, right column). The first row corresponds to the average on land, the second row corresponds to the average in irrigated areas, and the third row corresponds to the average in non-irrigated areas. The dashed lines correspond to a fitted polynomial curve via local fitting.**

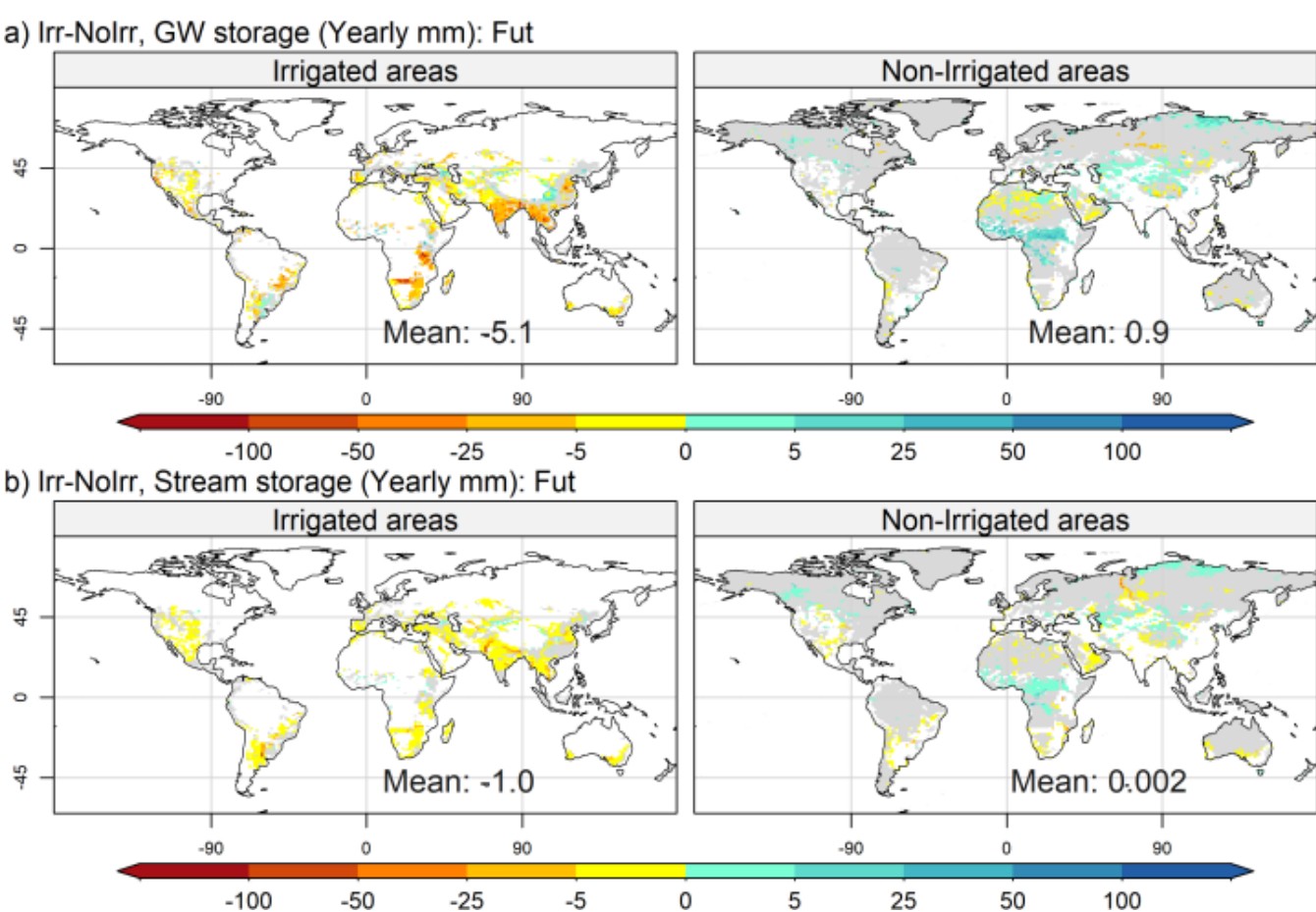

**Figure 8: Idem as in Figure 6 for groundwater storage (a) and stream storage (b). The areas in gray correspond to a p value less than 0.05 according to Student's t test.**

### 3.5 Joint effects of irrigation and climate change on river discharge

River discharge integrates all changes linked to climate change and irrigation at the basin scale. We decide to analyze the effects on yearly average values in the 50 largest river basins (Table S1), and we present monthly average discharge values of seven river basins that summarize the results obtained (Figure 9). We classify changes in river discharge into three classes, based on three variables: the average irrigated fraction during a period (historical or future), the effect of climate change on discharge, and the irrigation influence on discharge.

The first class corresponds to large river basins with heavy irrigation activities, i.e. with an average irrigated fraction higher than 1% during the historical or future period (illustrated by the Nile, Rio Grande, Indus and Ganges; Figure 9-a, b, c, and d). In these river basins, irrigation activities decrease discharge values throughout the year (decrease in the future due to irrigation ranges between -4 up to -51% of discharge) under both historical and future climate conditions, with no major changes in seasonality (except in Rio Grande). The impact of climate change increases the discharge values in the future, and the decrease

of discharge by irrigation is greater in the future than in the historical period, because the irrigated fraction increases, boosting the demand, and the increased water supply by river discharge allows irrigation withdrawals to follow the demand.

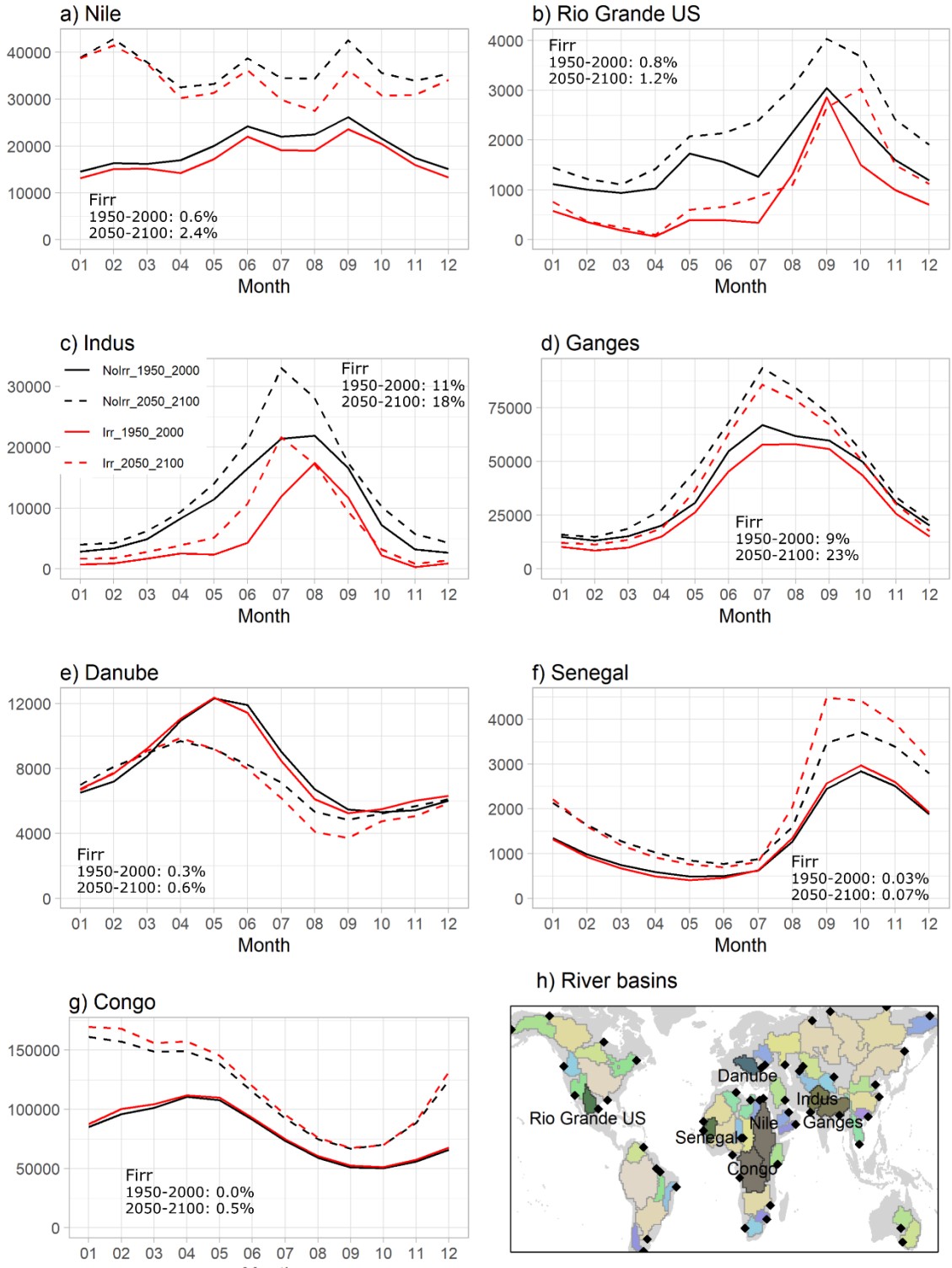

**Figure 9: Monthly multiyear values of discharge (m³/s) at the basin outlet in the chosen river basins (a-g) and the 50 river basins considered in the analysis (h), with highlights of the rivers presented in a-g.**

The second class is illustrated by the Danube River basin (Figure 9-e, as well as the Dnepr and Nelson river basins, Table S1), and corresponds to river basins with modest irrigation activities (i.e. average irrigated fraction lower than 1% during the historical or future period) and a decrease of river discharge with climate change (decrease due to climate change range from -1 up to -15% of discharge). Under the historical climate (solid lines) conditions, river discharge from the Irr simulation increases during the winter months compared with that from the NoIrr simulation (owing to a positive P change in the area),

but discharge values from the Irr simulation are lower than those from the NoIrr simulation during the summer (when water resources are used to sustain irrigation activities). Under the future climate, the positive difference during the winter is weaker, whereas the negative difference during the summer is greater (due to the regional decrease in P and increase in ET due to climate change and the increase in effective irrigation).

The third class also corresponds to river basins with modest irrigation activities (irrigated fraction lower than 1%) in both

periods, but a slightly positive influence of irrigation on discharge (increase in the future ranges between +3 to +12%). This class is illustrated by two African river basins: the Congo and Senegal river basins (Figure 9-f and g). Both basins show slightly higher discharge values for the Irr simulation than for the NoIrr simulation under the historical climate, and this positive difference is greater under the future climate scenario. The positive influence of irrigation under the historical climate is explained by the P increase in non-irrigated areas of both river basins in the Irr simulation. On the other hand, the greater

influence of irrigation under the future climate than under the historical climate can be explained by the negative impact of climate change on ET in the area (Figure 4) and the positive modulation of P in some areas of the Sahelian band (indicating more P in the Irr simulation in the future; Figure S4). Notably, positive modulation occurs during the wet season in both cases, and in the case of the Congo River basin, the modest irrigation expansion (the irrigated area increases from 0 to 0.5%) does not decrease discharge in the future, despite an increase in irrigation (Figure 1-d).

**3.6 Hydroclimate limits to irrigation growth**

We showed that irrigation has an important influence on the evolution of the water cycle and water resources in irrigated areas. This influence is modulated by the expansion of irrigated areas and can enhance the effects of climate change or reduce them, depending on local conditions. However, we have not yet analyzed whether climate change impacts can explain changes in irrigation, which could in turn modulate the influence of irrigation on the water cycle and water resources. To analyze this type

of interaction, we show a map of classes according to the joint changes in precipitation and irrigation in Figure 10-a and according to the joint changes in irrigation and GWS in Figure 10-b. We also show boxplots of average changes by class for key hydroclimate variables in Figure 11.

We can observe that the areas where precipitation decreases correspond to roughly one-fifth of the total irrigated areas (red and orange classes in Figure 10-a). In the orange zones, irrigation increases in response to a larger irrigated fraction, drier soil,

and higher net radiation (which increases the water demand, Figure 11), increasing irrigation as a trade-off between less

available water and more water demand. In the red zones, due to the shortage of water supply, irrigation cannot increase even if the irrigated surface increases in many zones (approximately half of the grid cells; Figure 11), the soil is drier, and the net radiation increases.

The difference in available water for irrigation explains the significant decrease in GWS in the orange zones due to the
combined effect of less P and greater water withdrawal, whereas in the red areas, the decrease in GWS is close to zero, as it is not possible to further increase water withdrawal, which is near the maximum under the historical climate. The effects of changes in irrigation also explain the differences in ET changes and LAI changes between the red and orange classes. Additionally, the decrease in the irrigated fraction, which might decrease the irrigation demand and effective irrigation, plays a minor role in the case of the red class.

Areas with increased precipitation account for four-fifths of the total irrigated area (green and blue classes in Figure 10-a). In the green areas, effective irrigation decreases, in part because the irrigated fraction decreases in many areas (slightly less than half of the grids, Figure 11; this class includes Iraq and the Mississippi River Basin where Firr decreases, Figure 10-b) but also because the climate change increase in soil moisture and net radiation is less important than in the blue class (for example, in some areas of China).

In the blue areas, irrigation increases (e.g., in China and India), partly because the irrigated fraction increases and because the increase in net radiation is more important (which increases water demand) despite the increase in soil moisture (which decreases water demand). These differences in SM and net radiation partly explain the differences in the changes in ET and LAI, which increase more in the areas classified as blue than in those classified as green. In addition, the areas classified as green could present a potential for the expansion of irrigated systems (if there is available space for agriculture) or even for
the use of rainfed systems if the future local climate allows it. But such policy decisions often depend on other socio-economic factors beyond environmental conditions (Mehta et al., 2024; Petit et al., 2017; Siebert et al., 2015).

With respect to the relationship with water resources, the blue and green areas generally show an increase in GWS, indicating an increase in water resources (Figure 11). However, in at least a quarter of the areas classified as blue, GWS decreases and irrigation increases despite the increase in P (i.e., some grid cells are classified as blue in Figure 10-a but as orange in Figure
10-b). This indicates that increased irrigation leads to a depletion of water resources, even under a wetter climate. In total, areas where changes in climate or irrigation intensity lead to water resources depletion (classes red and orange in Figure 10-b) constitute one third of the total irrigated area and include the intensively irrigated areas of northern India and Southeast Asia. A direct consequence is that areas classified as orange in Figure 10-b (less water storage while irrigation increases) may face more tension over water use because of overexploitation of water resources (in those areas where P increases) or due to
the trade-off between less rainfall and more water demand, while red areas in Figure 10-b may face a decline of irrigation activities due to the shortages of water supply.

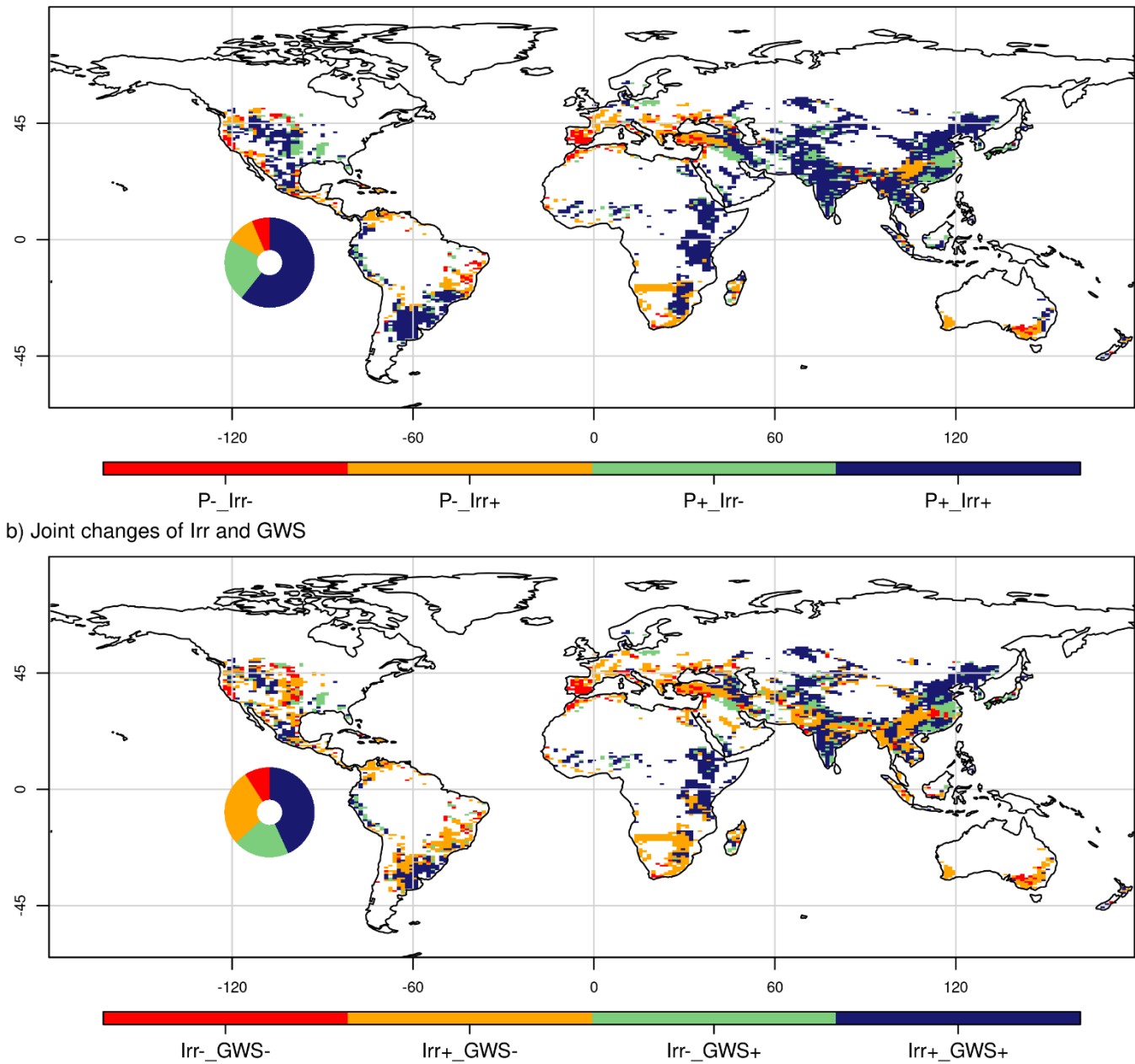

**Figure 10: Joint changes between the future (Fut, 2050-2100) and historical (Hist, 1950-2000) periods for precipitation (P) and irrigation (Irr) in irrigated areas (a) and for groundwater reservoirs (GWS) and irrigation (Irr) in irrigated areas (b). The symbols + and - indicate positive and negative changes, respectively. The insets indicate the fraction of irrigated area by class.**

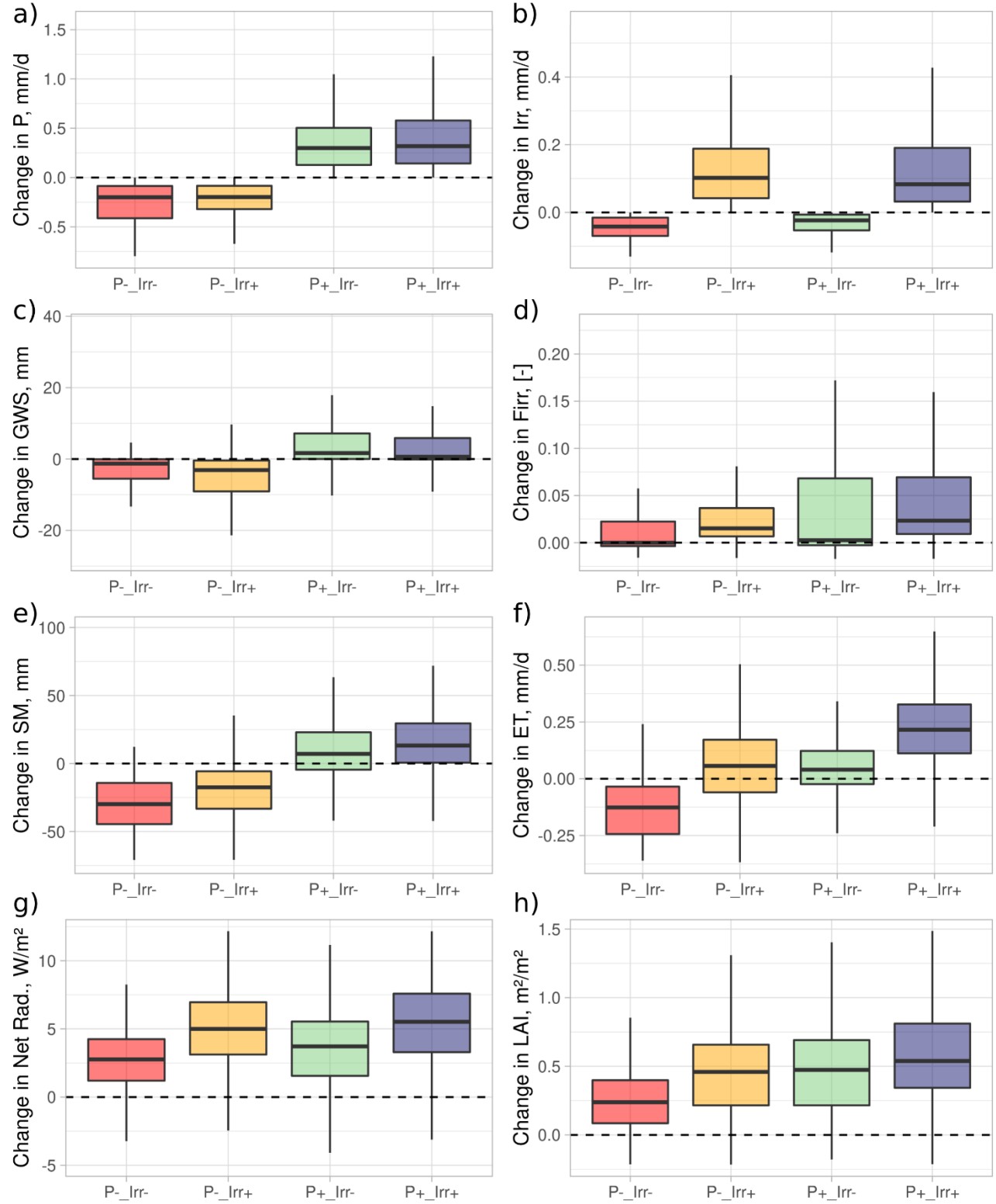

**Figure 11: Boxplots of average climate change impacts as simulated in the Irr simulation, in irrigated grid cells classified according to Figure 11-a (joint change in P and Irr), for precipitation (a), irrigation (b), GWS (c), irrigated fraction (d), SM (e), ET (f), net radiation (g), and LAI (h).**

## 4 Discussion

First of all, it is important to recall that our results are framed by several important uncertainties related to our modeling framework:

1. The use of SSP5-8.5 as a single radiative forcing scenario induces a strong warming and significant changes in precipitation. This scenario could be seen as the upper boundary of potential climate change impacts, but the magnitude and spatial distribution of these changes are uncertain (AR6, IPCC, 2021). . The direct impact of climate change on irrigation could be affected by this uncertainty, especially with regard to changes in precipitation patterns.

2. The ocean component in our simulation is prescribed, while there is evidence that ocean-atmosphere interactions modulates irrigation impacts (Krakauer et al., 2016).

3. Uncertainty due to internal variability, i.e. the natural variations in climate due to interaction of different components of the Earth System, is not considered, since we use a single simulation instead of an initial-condition ensemble (Schwarzwald and Lenssen, 2022).

4. Some specific processes report uncertainty as well, for instance the response of convective storms to warming (Lepore et al., 2021), the magnitude of some energy balance components, especially latent heat flux (Wild, 2020), or the representation of local breeze circulation at the subgrid scale (Lunel et al., 2024).

5. The evolution of SM-atmosphere coupling in climate projections is uncertain (Qiao et al., 2023; Zhou et al., 2021), especially in regions identified as transition zones with strong land-atmosphere coupling (Koster et al., 2006; Seneviratne et al., 2013) but this uncertainty is not considered since no multi-model ensemble is used.

6. The misrepresentation of subgrid variability in the land surface-atmosphere coupling (de Vrese et al., 2016b), specifically on the energy budget computed by IPSL-CM6, could have an impact on irrigation demand, on atmospheric feedbacks and on irrigation efficiency (de Vrese and Hagemann, 2018).

These uncertainties quantitatively influence the entire simulated water cycle, so that our results are model-specific and must be confirmed using other land-atmosphere models and available observations.

Our results are also strongly dependent on the shortcomings of ORCHIDEE's representation of crop irrigation and its water sources, some of these shortcomings shared with other LSMs (McDermid et al., 2023):

1. The ORCHIDEE LSM lacks a parameterization of deep, non-renewable GW, and the map of water access infrastructure is fixed to year 2000-conditions (Arboleda-Obando et al., 2024). Also, river water adduction is deactivated in the Irr simulations due to the coarse resolution These three shortcomings may reduce the water

supply for irrigation, thus limiting the simulated irrigation withdrawal. They call for a massive effort to better

account for human water management.

2.    Our irrigation parameterization uses flood irrigation as the only irrigation technique worldwide and overlooks local irrigation practices (Arboleda-Obando et al., 2024). Model choices (such as the use of global parameters, simplified irrigation rules, and lack of specific crop phenology) necessarily impact the regional magnitude of water demand and thus the evolution of irrigation. Here again, these shortcomings highlight the need for more complex human

water use modules in LSMs (Taranu et al., 2024; Yao et al., 2022).

3.    The irrigated crops are also very simplified in our model: our single C3 and C4 crop types follow a natural phenology, without harvest, which probably leads to overestimate the simulated irrigation at the end of the growing season; in contrast, multiple cropping and paddy fields are overlooked, leading to underestimate the simulated irrigation (Arboleda-Obando et al., 2024). The overall effect of these shortcomings are difficult to assess

quantitatively, and they are not exhaustive.

With these limitations in mind, we analyze how our findings usefully complement those of four important articles on irrigation related trends. Firstly, our conclusions regarding the influence of irrigation on ET and P match those of Cook et al. (2020), but with a more solid framework in our case. The simulations of Cook et al. (2020) benefit from an ensemble of multiple simulations thus better assessing internal variability, but all the simulations include irrigation, so the influence of irrigation is

approximated from the comparison of irrigated and nearby non-irrigated areas, where the effect of irrigation on ET and P is assumed to be zero. Our results, albeit based on one member only, show that irrigation significantly increases P and ET in many areas surrounding irrigated regions, so the influence of irrigation estimated by Cook et al. (2020) on these two variables were likely underestimated.

Analyzing the atmospheric impact of irrigation in more remote regions is challenging and was not comprehensively addressed

in this study. De Vrese et al. (2016a) tackled this issue by carefully studying how wind patterns are modified by irrigation in coupled simulations of historical climate, and showed that irrigation in India leads to increase P in East Africa and Central Asia, owing to water vapor advection and disturbances in the Asian Monsoon. Our results focused on the effects of irrigation on water resources, and therefore the effects on atmospheric circulation were not analyzed in detail, although similar processes may exist in our simulations. A better understanding of the effects of irrigation on precipitation in remote areas must define

what a "remote area" means, then analyze the relationship between irrigation (taking into account the intensity and spatial distribution of irrigation), and changes in atmospheric circulation and moisture transport in a given climate (either historical or future). This analysis can be more robustly complemented by the use of inverse trajectory methods (Wei et al., 2013), to estimate the contribution of irrigation to total precipitation, and by the use of water tagging, to track the origin of water vapor and continental recycling (Risi et al., 2013) and study the changes induced by irrigation.

Our conclusions regarding irrigation growth at global scale for the period 1950-2100 are in line with irrigation projections based on agronomic schemes for the 21st century (Busschaert et al., 2022; Hanasaki et al., 2013; Wada et al., 2013). These conclusions, however, only partially agree with the results of Khan et al. (2023), which show a decrease of irrigation at global

scale for scenario SSP5-8.5. This divergence is partly due to a decreasing trend in irrigated areas worldwide in Khan et al. (2023), which is opposite to the positive trend prescribed by LUHv2. Results from Khan et al. (2023) rely on an integrated

assessment model, GCAM (Graham et al., 2020) which focuses on the integration of various socio-economic processes and assumptions, with a prescribed, non-interactive, future climate scenario. Estimates of irrigated fractions within GCAM are based on lumped values of water supply and water demand at the coarse scale of large river basins (Chen et al., 2020), although these terms usually display strong irrigation and water resources heterogeneities in the real world. This calls for two perspectives:  firstly, to explore the effect of varied scenarios of irrigated areas on irrigation withdrawal in climate models;

secondly, to explore the inclusion of some socio-economic factors that control the evolution of irrigation and could improve the representation of irrigation in LSMs, but are currently missing.

Finally, our findings on the hydroclimatic limits of irrigation activities bring a new perspective to the question of irrigation sustainability. Our conclusions largely match those of Mehta et al. (2024), despite a different methodology and period of work. Based on observed historical datasets, Mehta et al. (2024) show that many blue water-stressed regions exhibit an unsustainable

increase of irrigated areas in the early 21st century Based on a long-term simulation that considers changes in hydroclimatic conditions and changes in irrigated areas, we also identify areas that could undergo unsustainable irrigation water withdrawals at the end of the 21st century due to changes in the blue-water stress. Regions where our results differ from those of Mehta et al. (2024) include irrigation hotspots such as the Mediterranean basin, southern South America, and Southeast Asia, which we identify as areas of unsustainable irrigation in the future as a result of a more arid climate (e.g. the Iberian peninsula) or

overexploitation of water resources (e.g. southern South America and Southeast Asia). These results call to further explore the sustainability of future irrigation scenarios within GHMs and LSMs, an important step for supporting the design of adaptation policies at the regional scale for the water–food nexus.

**5 Conclusions**

We explored the joint evolution of irrigation activities, the water cycle and water resources under the SSP5-RCP-8.5 climate

change scenario, a scenario that could be seen as the upper boundary of potential climate change impacts. Our results suggest that climate change and irrigation expansion will continue increasing global irrigation withdrawal in the future. The spatial distribution of irrigation change is heterogeneous and depends on regional changes in climate and irrigated areas. The enhancement in ET induced by irrigation will increase in irrigated areas and over land in the future. ET will also increase in non-irrigated areas near irrigated zones owing to an increase in precipitation, but the influence of irrigation remains similar

under historical and future climate. Irrigation increases precipitation (with local exceptions), and the influence of irrigation on precipitation remains similar under historical and future climate as well.

Historical depletion of water resources by irrigation will continue in the future, but our simulations reveal no major effect of irrigation on river discharge seasonality under historical and future climate. In irrigated areas, climate change increases average water resources (surface and groundwater), but irrigation activities can modulate the speed of change. In some non-irrigated

areas near irrigated zones, the increase in precipitation due to irrigation increases water resources under historic and future climate (for instance, the Sahelian band and Central Asia), but the average irrigation influence in areas with no irrigation remains weak throughout the simulation period. Importantly, we identified the areas in our simulation where irrigation could be limited by future hydroclimate conditions (red class in our results) and the areas where irrigation could increase tensions over the use of water resources (orange class in our results), which represent roughly one-third of the total irrigated areas (the Mediterranean basin, Australia, California, Southeast Asia). Conversely, we determined areas where irrigation activities could intensify or even partially convert to rainfed systems.

Eventually, our results underline the importance of including irrigation in climate change projections. It allows us to assess the influence of future irrigation on water resources. Also, it helps to understand the complex feedback loops between irrigation, water resources and future climate at global and regional scale, which is crucial for supporting the design of adaptation policies of the water-food nexus in intensively irrigated regions. We remark that the description of irrigation in LSMs within ESMs presents large uncertainties related to its biophysical and socio-economic drivers. This calls for enhanced multi-model studies, as initiated within the IRRMIP framework (Yao et al., 2024), but it also calls for an important interdisciplinary effort to produce contrasted yet plausible scenarios of future irrigated areas and irrigation practices, ideally accounting for the feedback from non-agricultural water demand.

**Author contribution**

AD, FC and PFAO designed the study, AD, PFAO and JG run the simulations; PFAO wrote the first draft. All the authors contributed to interpreting the results, discussing the findings and improving the final version of the paper.

**Code availability**

The version of the ORCHIDEE LSM used for this study corresponds to tag 2.2, revision 7709 (Arboleda-Obando et al., 2023), and is freely available from https://forge.ipsl.jussieu.fr/orchidee/log/branches/ORCHIDEE_2_2/ , It is provided under a CeCILL-C license (French equivalent to the LGPL license).

**Competing interests**

The contact author has declared that none of the authors has any competing interests.

**Acknowledgement**

This work has received support from Belmont Forum, BLUEGEM project (grant no. ANR-21-SOIL-0001). The simulations were done and stored using the IDRIS computational facility (Institut du Développement et des Ressources en Informatique Scientifique, CNRS, France) on the supercomputer Jean Zay CSL, under the allocation 2022[AD010113599R1].

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
