# Peer review of "Joint evolution of irrigation, the water cycle and water resources under a strong climate change scenario from 1950 to 2100 in the IPSL-CM6"

_Earth System Dynamics, 2024_

## Author Comment (AC1)

- **RC1**: 'Comment on esd-2024-41', Anonymous Referee #1, 14 Apr 2025

In this manuscript, the authors use IPSL-CM6 simulations to the study the effects of irrigation on regional water cycle and water resources. They conduct simulations with and without irrigation from 1950–2100 under historical and SSP5-8.5 scenario. They find that irrigation expansion and climate change will intensify water use and water stress, while the spatial distribution of these changes vary across regions.

This is a comprehensive assessment, highlighting the need to include irrigation in models for climate change analyses, especially in regions with extensive irrigation. The methodology was carefully developed and overall, the manuscript is well-structured. However, it requires effort on two aspects: first, the interpretation of figures and results in the manuscript is not always accurate, and second, the explanations provided lack the depth and rigor needed to support the conclusions. The comments below may help address these concerns; however, the authors need to provide a thorough explanation of their results, considering both the model limitations and how user choices affected them.

We thank the anonymous reviewers 1 for the time he spent reading and commenting on our paper. Below, we provide a point-by-point response to these comments. Sentences from the original submitted manuscript are presented in italic, while the proposition to respond to the observations are presented in bold. Lines correspond to the original manuscript.

1. Ln 64: Please expand all model abbreviations (LMDZ, LMDZ6A, LMDZOR, ORCHIDEE, STOMATE, etc.) in this section. Is LMDZOR the coupled LMDZ and ORCHIDEE model?

We thank the reviewer 1 for his observations on the paper.

Yes, LMDZOR corresponds to the coupled LMDZ and ORCHIDEE model. We expand the model's abbreviation in the manuscript so it is more clear for the reader.

In line 65:

*We used the LMDZOR model (Cheruy et al., 2020), which involves the coupling of the* **atmosphere** *and* **land** *components of the IPSL-CM,* **respectively the LMDZ (Laboratoire de Météorologie Dynamique with Zoom capacity) and ORCHIDEE (Organising Carbon and Hydrology In Dynamic Ecosystems) models. We briefly describe each component.** *This version uses the LMDZ6A atmospheric model (Hourdin et al., 2020;* **Sadourny and Laval, 1984***) embedded in IPSL-CM6A-LR (Boucher et al., 2020)*

In line 80:

**The Organising Carbon and Hydrology In Dynamic Ecosystems,** ORCHIDEE land surface model

In line 91:

*(...) and **carbon fluxes** and plant phenology are controlled by the STOMATE (Saclay Toulouse Orsay Model for the Analysis of Terrestrial Ecosystems; Krinner et al., 2005) module (...)*

2. Ln 92: Can you briefly describe here (or in the Discussion section when limitations are discussed), how absence of a crop phenology module affects the results? Would this result in overestimation of water use if irrigation is applied year-round?

We didn't use a specific crop phenology module owing to a lack of ubiquitous parameters at the global scale. It means that C3 and C4 crops are assumed to have the same phenology as natural grasslands but with higher carboxylation rates.

As we discussed in the presentation of the irrigation module (Arboleda et al., 2024), the use of a simplified representation of crop phenology may have an effect on the timing (the moment of applying irrigation) and the volume of irrigated data. It could induce an overestimation of water demand and ultimately water use (if available water offer).

We agree that uncertainties of the crop representation are important, we propose to add a further explanation in line 92:

*Note that no specific crop phenology module was used, following Arboleda-Obando et al. (2024a), **owing to a lack of ubiquitous parameters at the global scale. It means that C3 and C4 crops are assumed to have the same phenology as natural grasslands but with higher carboxylation rates. Remark that this simplification may have an impact on the timing and volume of total water withdrawal, by inducing an overestimation of water demand and ultimately water use (if there is available water offer; Arboleda-Obando et al., 2024a).***

3. Ln 113: Please provide a justification for the moisture deficit factor (0.9). Same for other user-defined parameters (max. irrigation per hour, root zone).

Are these parameters global or can they be defined by region, crop type, season, etc.?

Also, did you conduct sensitivity analysis of these parameters on model outputs? In subsequent sections, it is brought up that model choices effect results (e.g., Ln 317), so it is important to quantify model sensitivity to these parameters.

Irrigation parameters are assumed to be uniform in all irrigated areas, with no distinction by region, crop type, season, etc. Given this simplification, the use of a sensitivity analysis and a parameter tuning is important. We applied an extensive evaluation of the irrigation module and its effects in the ORCHIDEE LSM that we presented in Arboleda et al., 2024, using offline simulations. The evaluation includes a simple sensitivity analysis and a simple parameter tuning.

The sensitivity analysis showed that the moisture deficit factor is the most sensitive parameter, followed by the maximum irrigation per hour and finally the root zone definition. Given this information, we decided to fit simulated irrigation to global observed data by tuning the SM deficit factor. A value of 0.9 was a compromise between total global volume

and local biases. For the other parameter values, which are less sensitive to changes, we decided to use "reference values" that showed good results.

While we consider that describing in detail these results is not worth it in the manuscript, we agree with the reviewer that it is important to clarify why we use these parameter values, and refer to the paper that shows the sensitivity analysis and the parameter tuning. We propose to add the following sentence in line 111:

*The irrigation scheme used here was tested and evaluated in Arboleda-Obando et al. (2024a) at the global scale.* **This work included a sensitivity analysis to understand the effect of each parameter on irrigation volume and evaporation increase, leading to choosing a set of globally homogenous parameters enabling a good match between simulated and reported values of global irrigation withdrawal.**

Also, we added a new paragraph to explain the effect of the simplified representation of crops in the model, see response to observation 2 from reviewer 2.

4. Ln 90 mentions that 15 PFTs are represented with different parameter values – is this referring to the irrigation scheme parameters?

No, it refers to the plant phenology as represented in the STOMATE module, i.e. to parameter values that are used to simulate photosynthesis and carbon allocation. As mentioned in the document, STOMATE controls the carbon fluxes and the plant phenology. We understand that the phrase may be confusing, so we propose to add a sentence in line 90:

*Vegetation is represented by 15 plant functional types (PFTs, including bare soil), each with different parameter values* **to simulate photosynthesis and carbon allocation; carbon fluxes** *and plant phenology are controlled by the STOMATE (Saclay Toulouse Orsay Model for the Analysis of Terrestrial Ecosystems) module, which computes the evolution of the leaf area index (LAI) (Krinner et al., 2005).*

5. Ln 116: Where all three natural reservoirs are accessible, does the model prioritize one reservoir or withdraws equally from them?

The model prioritizes the reservoirs according to facility for accessing the natural reservoir, as represented by the presence of irrigation equipment for groundwater use and surface water use. This information is prescribed to the model by a map of the area equipped for surface and groundwater use. For instance, if we are located in an irrigated grid cell where there is no pumping infrastructure to access groundwater, all water supply will come from the surface (overland and river reservoirs).

To clarify the priority process, we propose to add a sentence in line 119:

*(...) and the facility for accessing the natural reservoir as represented by a prescribed map of irrigated areas that are equipped for surface or groundwater use (Siebert et al., 2010).* **The facility of water access helps to prioritize one reservoir over others; for instance, in a gridcell with irrigation demand and groundwater availability but no groundwater access infrastructure, all the water supply will come from the surface reservoirs (overland and stream). More details can be found in Arboleda et al (2024a).**

6. Ln 121: Would disabling adduction from neighboring grid cells result in underestimation of irrigation water availability? I understand the need for this in coarse resolution simulations, but perhaps the authors can expand on this in the discussion/limitations section.

In Arboleda et al., (2024), we show that disabling adduction may have an important decrease of simulated irrigation. Adduction is important in some basins with extensive irrigation with water from large rivers, because in those basins there are important areas equipped for irrigation in neighboring grid-cells to those large-river grid cells. Two cases are the Nile river basin and the Indus river basin.

In the case of a coarser resolution, these cases should be less frequent and its impact on the total irrigation volume should be less important, as suggested by the total irrigation volume in the online simulation (around 2300 km$^3$ in year 2000), which is close to the result from the offline simulation in Arboleda et al., (2024). Despite this, we understand that the adduction representation that we use is rather crude and could be improved by including human water management and dams operation.

We agree then that this issue is worth discussing in the discussion section. We add a sentence in line 408:

*The ORCHIDEE LSM lacks a parameterization of deep, non-renewable GW, and the map of water access infrastructure is fixed to year 2000-conditions (Arboleda et al., 2024).* ***Also, river water adduction is deactivated in the Irr simulations due to the coarse resolution These three shortcomings*** *may reduce the water supply for irrigation, thus limiting the simulated irrigation withdrawal.* ***They call for a massive effort to better account for human water management.***

7. Ln 133: What was the reason for choosing the 5-8.5 scenario? The stronger warming will amplify the differences – did you compare the results with a middle-of-the-road scenario (e.g., 2-4.5)? I understand this is in the title ("strong climate change scenario"), so new simulations are not expected, but please highlight the effects of this choice both in Discussion (Ln 393) and in the Conclusions (Ln 446). This scenario gives the upper bounds of potential climate impacts, providing the worst-case scenario, so statements like "water depletion due to irrigation is more intense in the future than historical period…" need to be carefully presented.

We agree with the reviewer that the reasons for choosing the SSP5-8.5 scenario are not clearly stated. The main reason is to have a strong climate signal, not only on global warming but also on precipitation changes, and the use of this scenario is indeed an upper boundary of potential climate change impacts. We did not compare with a middle-of-the-road scenario.

We propose to reinforce the idea that the simulation is using a scenario with the strongest climate change impact. We proposed to add a sentence in line 134:

*The radiative forcing is prescribed using historical (1950-2014) and SSP5-RCP8.5 (2015-2100) datasets from ScenarioMIP (Tebaldi et al., 2021).* ***The use of scenario SSP5-RCP8.5 could be seen as the upper boundary of potential climate change impacts and results in a strong global warming and important changes in precipitation.***

In line 393 we propose to add a sentence in the same sense:

*The use of SSP5-8.5 as a single radiative forcing scenario induces a strong warming and significant changes in precipitation.* **This scenario could be seen as the upper boundary of potential climate change impacts**, *but the magnitude and spatial distribution of these changes are uncertain (AR6, IPCC, 2021).*

Finally, we propose to add a sentence in line 447:

*We explored the joint evolution of irrigation activities, the water cycle and water resources under the SSP5-RCP-8.5 climate change scenario*, **a scenario that could be seen as the upper boundary of potential climate change impacts**.

8. Ln 176: The explanation of modulation may not be clear to the reader and needs to be parsed out. "Climate change by irrigation" and "irrigation by climate change" lack clarification. Do you mean how climate change alters the effects of irrigation and how irrigation alters the response to climate change?

Thanks to the reviewer for this observation. Yes, when we talk about modulation, we are referring to additional changes on climate change effects that are induced by irrigation, and vice versa, to additional changes on irrigation effects induced by climate change.

We proposed to clarify the term modulation on line 177:

*This modulation can be described* **as the effect of irrigation on the alterations produced by climate change**, *i.e., [Irr(Fut) - Irr(Hist)] - [NoIrr(Fut) - NoIrr(Hist)], but this modulation is equivalent to* **the effect of climate change on the alterations produced by irrigation**, *i.e., [Irr(Fut) - NoIrr(Fut)] - [Irr(Hist) - NoIrr(Hist)].*

9. Ln 180: What does "hillslope flow" mean here? Earlier it was stated that there is no communication between neighboring grid cells (Ln 122), so how does hillslope effect come into play?

Thanks for mentioning this sentence, which is leading to confusion. In this study, there is no communication between neighboring grid cells nor representation of hillslope flows at the subgrid scale. But we use the concept of modulation from another study that used a special version of the ORCHIDEE Land Surface Model with a novel description of the effect of hillslope at subgrid scale (Arboleda Obando et al., 2022) . In this version of ORCHIDEE with hillslope flow representation, the groundwater and overland reservoirs can interact with the surface through a "lowland fraction" that buffers the flow between the former reservoirs and the river system.

The concept of modulation is useful because it allows us to understand the dynamic interaction between any processus in the continental component of the climate model, and a changing climate. We mention here that study, to underline this flexibility.

To clarify the reason to cite the study on hillslope flow and climate change, we propose to add a sentence in line 181:

*This modulation term is similar to the one introduced in Arboleda Obando et al. (2022) to characterize the coupled changes of climate change and hillslope flow on the basis of trends.* **While in this study there is no representation of hillslope flows, the concept of modulation is still useful to understand the dynamic interaction between a processus in the continental component of the climate model, and a changing climate.**

10. Ln 186: The 76% increase is based on the mean of future (2100 – 1950) and historical (1950 - 2014) periods?

In section 2.4, first point, we define the future period in our study as 2050-2100, and historical period as 1950-2000. We use this convention all along the study, and we use the names future (or fut) and historical (or hist) to avoid repeating the year's period every time.

To avoid confusion, we propose to modify the sentence in the beginning of section 3, line 186:

**In the SSP5-RCP8.5 framework,** *irrigation continues to increase throughout the simulation period (+76% in the future*, **for years 2050-2100;** *compared with the historical period*, **1950-2000**), (...)

11. Ln 187: The interpretation of Figure 2b is not clear, I do not think it highlights "no major change in seasonality", especially for DJF and MAM seasons. Please explain this further.

We agree that the sentence doesn't explain what Figure 2b shows. We propose to change the sentence so it shows that there is a seasonal increase for all seasons, but these changes do not alter the relative importance of every season on the total irrigation volume.

*In the SSP5-RCP8.5 framework, irrigation continues to increase throughout the simulation period (+76% in the future, for years 2050-2100; compared with the historical period, 1950-2000).* **At the seasonal scale, global irrigation increases for all four seasons, and the relative weight of each season is not markedly changed throughout the simulation period, although the weight of DJF increases slightly to the expense of MAM; JJA remains** *the main irrigation season* (see Figure 2-b).

12. Ln 197: I do not see decrease in irrigation in northern India, it seems upper Indus has some decline. It would be helpful to add watershed outlines in Fig. 3 for better reference of domains.

We agree that decrease is not general in all of northern India, but in the upper Indus river basin and some areas of Northeastern India, in the Ganges river basin. We propose to change the area listed so it is more precise, line 197:

*However, other areas are less irrigated despite the expansion of irrigated areas (see areas* **in the upper Indus river basin, in the lower Ganges river basin** *and in the Iberian Peninsula).*

Regarding the observation on basins' contours, we prefer to leave the figures as they are, considering they already show a lot of information. But we decided to change the colorbars so it is more intuitive (see observation 10 from reviewer 2) and to include this figure into a composite figure with Fig. 1 (see observation 26 of reviewer 1)

13. Ln 199: Can you elucidate what other climate factors can explain the evolution of irrigation and how? Could the differences between irrigated areas and irrigation stem from model choices (global parameters, simplified schemes, no crop phenology)? There needs to be a more careful discussion on what differences can be attributed to climate driven vs model assumptions.

In this section (3.1 " Evolution of irrigation under a changing climate") we show that the increase of the irrigated area in the SSP5 scenario is not enough to explain an increase of irrigation. It could mean that other climate factors can help to explain the evolution of irrigation. We then explore the relationship between other climate factors and evolution of irrigation in section 3.6 "Hydroclimate limits to irrigation growth". We show that there is a complex relationship between future climate and evolution of green and blue water, changes in the irrigated area, and evolution irrigation.

In any case, we understand that our results are affected by uncertainties due to model choices and model limitations. The uncertainty could affect the magnitude of irrigation changes. But so far, we are unable to elucidate the uncertainty driven by model assumptions. On the other hand, the IRRMIP project (irrigation model intercomparison project in which we participate) shows that the differences in the historical evolution of irrigation between models can be important (Yi Yao et al., 2024).

We agree that the interpretation of results must consider the uncertainty due to model choices, and should be part of the discussion. We propose to add two main changes.

First, in line 199:

*This means that climate factors could contribute to explain the evolution of irrigation.* ***But it should be noted that model choices, such as the use of global parameters, a simplified representation of irrigation and the lack of crop phenology, can influence the magnitude or even the direction of water demand and thus the evolution of irrigation.***

Second, in section 4 "Discussion", line 412:

*Our irrigation parameterization uses flood irrigation as the only irrigation technique worldwide and overlooks local irrigation practices (Arboleda et al., 2024).* ***Model choices (such as the use of global parameters, simplified irrigation rules, and lack of specific crop phenology) necessarily impact the regional magnitude of water demand and thus the evolution of irrigation. Here again, these*** *shortcomings highlight the need for more complex human water use modules in LSMs (Yao et al., 2022; Taranu et al., 2024).*

14. Ln 209: Precipitation increases in both irrigated and non-irrigated areas, so it may not be due to the influence of irrigation alone.

Here we are presenting the average values for land, irrigated regions and non-irrigated regions. The single difference between simulations is that the irrigation is activated, or not. It indicates that an increase of irrigation can be directly or indirectly related to irrigation.

In section 3.4, we show that the increase of P in non-irrigated regions is located in areas next to irrigation hotspots. It indicates that irrigation can induce changes in non-irrigated areas, that

are next to irrigation hotspots. While our results may be affected by internal variability, our results are robust enough to show a link between irrigation and increase of precipitation in non-irrigated areas that are not far away from irrigation hotspots.

To clarify the idea that we are identifying here the average effect of irrigation but that we assess the spatial distribution in a further section, we propose to add a sentence in line 208:

*Table 1 shows the average values from **both** Irr simulations of 10 key hydroclimate variables, the influence of irrigation and climate change impact on those variables, and the modulation for land, irrigated areas and non-irrigated areas. **In section 3.4 we assess the spatial distribution of the influence of irrigation for P, ET and water resources.** Climate change accelerates the water cycle, warms the air, and increases net radiation. We observe that the influence of irrigation **(i.e. the difference between the NoIrr and the Irr simulations)** increases the average **land** values of ET, precipitation (P), runoff (R), and LAI, while it depletes water storage in irrigated areas, i.e., groundwater storage (GWS) and stream storage (Stream S), but increases water storage in non-irrigated zones. The influence of irrigation on total water storage (TWS) is positive, which is partially due to an increase in soil moisture (SM).*

15. Are irrigated and non-irrigated areas identified as the cumulative shaded and grey grid cells in Figure 1a, respectively?

Yes, it is as the reviewer describes. We propose to add a clarification in line 167-168:

*We considered that a grid cell belongs to the "irrigated areas" if the average irrigated fraction from 1950-2100 was different from zero **(see shaded grid cells in Figure 1a and b)** or to the "non-irrigated areas" if the average irrigated fraction was equal to zero for the same period (see white grid cells in Figure 1a and b).*

16. Section 3.3: Fig. 4 is for ET and R, not for precipitation. Please revise this section and the figure.

Note that figure 4 in the original manuscript is now figure 3.

We thank the reviewer for this typo. Figure 3 corresponds to precipitation and air temperature, but the title of Fig. 3-a and b are wrong. The section correctly refers to P and Tas in section 3.3. We will correct the titles in the new version of the manuscript.

17. Ln 240: Please clarify this statement: "Additionally, stream reservoirs tend to show the strongest changes in the grid cells containing the largest rivers."

In Figure 4-d, we can observe that main changes in stream reservoirs are located in grid cells that contain the largest rivers. It is visible in the Amazon river, the Nile river, the Congo river, and the Indus river. The reason is that any increase (note that a decrease is also possible) in large river basins propagates across the river drainage network and concentrates in the grid cells with large rivers within.

We propose to explain this in line 240:

*Additionally, stream reservoirs tend to show the strongest changes in the grid cells containing the largest rivers, **since changes in the stream water budget of any grid cell propagates***

*along the river network, and accumulate in the grid cells with large upstream areas (e.g. Amazon river, Nile river, Congo river and Indus river).*

18. Ln 260: Again, is this referring to cumulative irrigated and non-irrigated areas? Do non-irrigated areas also include regions in the North Pole?

Yes, it refers to the cumulative irrigated and non-irrigated areas. In addition, mean land refers in this study to all continental land except Greenland and Antarctica.

We consider that reference to land is not clear enough, so we propose to add an explicit clarification in line 167:

*We focused on average changes over land, in irrigated areas and in non-irrigated areas, for irrigation as well as important land and atmospheric variables related to the water cycle. Land refers here to all continental areas except Greenland and Antarctica, which are not represented by the ORCHIDE LSM. We considered that a grid cell belongs to the "irrigated areas" (...).*

And to remember in line 260 this definition:

*The evolution of the ET and P yearly average rates over land, for irrigated areas and non-irrigated areas, as defined in section 2.4, is shown in Figure 5.*

19. Ln 261 - 265: Which panels are these statements referring to? Please add respective panel with each statement.

Note that figure 6 in the original manuscript is now figure 5.

We thank the reviewer for remarking this. We propose to add statements so is clear to which panel are we referring to:

*For ET, the NoIrr simulation shows a decreasing trend in irrigated areas during the 1950-2025 period that is not present in the Irr simulation (see Fig. 5a, second row). Additionally, the changes in ET observed over land (Fig. 5a, first row) are driven by changes in irrigated areas, as the ET values in non-irrigated areas are similar for both simulations (Fig. 5a, third row). Finally, we observe that the increase in ET in irrigated areas after 2025 is faster in the Irr simulation than in the NoIrr simulation, even though irrigation expansion stops by 2040 (Figure 2-a). In the case of P, irrigation activities increase the yearly average values over land (Fig. 5b, first row) and in irrigated areas (Fig. 5b, second row), but there is no major influence on the evolution over time.*

20. Ln 265: This statement needs to be parsed out. What does "no major influence on the evolution over time" mean? There is a positive trend 1950 – 2100, which is also present in the non-irrigated areas. So, there are other factors driving changes in P.

We agree that this statement needs clarification.

We state that irrigation doesn't have a major influence on the evolution of P, as both simulations (NoIrr and Irr) depict a similar positive trend over time. But we also state that this positive trend that we observe in Fig. 5b (new notation) is explained by all the other factors considered in our simulation, including climate change and land use change.

To clarify these two ideas, we propose to complete the sentence in line 265:

*(...) but there is no major influence on the evolution over time **(both NoIrr and Irr simulations show a similar positive trend over the period). This means that the positive trend in precipitation over land (Fig. 5b, first row) is driven by other forcings, i.e., climate change and land use and land cover change.***

21. Ln 288: Which panel of Figure 8?

Note that figure 8 in the original manuscript is now figure 7.

We agree that it is not clear which panel we are referring to in this section. We propose to change the sentence so it is more clear, after line 288:

*Time series of water storage in groundwater (which represents shallow aquifers, **see Fig. 7a**) and stream (which represents large rivers**, see Fig. 7b**) reservoirs **show important differences between NoIrr and Irr simulations. These differences are explained by** complex interactions between irrigation activities, climate conditions, and water resources (see **differences over land,** Fig. 7 **first row; irrigated areas, Fig. 7 second row; and non-irrigated areas, third row**) **that we pass to show**. The impact of climate change induces a positive trend in water storage **(Fig.7-a and b, first row)**, whereas irrigation decreases the average GWS and Stream S in irrigated areas **(Fig.7-a and b, second row)** and slightly increases the GWS and Stream S in non-irrigated areas **(Fig.7-a and b, third row)**.*

22. Ln 292: I am not sure about this statement: "…whereas the increase in water resources in non-irrigated areas in the Irr simulation is explained by the increase in precipitation in those areas near irrigated zones." This statement is somewhat misleading, implying that the changes in non-irrigated areas are influences by irrigated regions. In Fig. 7d (bottom right panel), there are P changes farther away from the irrigated areas (e.g., the Russian Tundra).

Also, please explain the changes in the ET (Fig. 7, top right panel) over Russia.

This statement is not misleading, we observe an increase of P, especially in areas around irrigation hotspots, that explains the increase of water resources. In farther away regions we also detect P increases, but we are more cautious about the direct mechanism (see observation 7 to reviewer 2). We explain this increase by a direct relationship with nearby irrigation. We propose to add a sentence in line 292 (Note that figure 7 in the original manuscript is now figure 6):

*The negative effects in irrigated areas are explained by direct water use to sustain irrigation activities, whereas the increase in water resources in non-irrigated areas in the Irr simulation is explained by **the fact that irrigation increases precipitation remotely, in particular around irrigated zones (Fig. 6b, right panel).***

Also, we extend the discussion on the effect of irrigation on P in remote areas, please see observation 7 from reviewer 2.

Regarding changes in the ET, there is a decrease of net radiation in the same area (see Fig. S10), probably linked to a decrease of shortwave radiation, but so far we cannot elucidate if it is due to a change in regional climate or if it is related to internal variability.

*The joint increase of ET and P in non-irrigated areas around irrigated areas reveals a remote impact of irrigation linked to atmospheric transport of moisture from irrigated to surrounding areas, which supports higher P and therefore higher ET in non-irrigated areas,* such as the Sahelian band and Central Asia. *Changes in P farther away from irrigated areas are rare and may result from various atmospheric processes in a generally more humid atmosphere.*

23. Ln 317: Percent values of what?

We think that this sentence does not describe the main idea of our analysis. We propose to change the sentence, so the main idea is more clear:

*(...). The impact of climate change increases the discharge values in the future,* **and the decrease of discharge by** *irrigation is greater in the future than in the historical period,* **because the irrigated fraction increases, boosting the demand, and the increased water supply by river discharge allows irrigation withdrawals to follow the demand**.

24. Ln 323: It seems the difference between irrigation and no-irrigation is trivial here. Please also refer to a later comment regarding figure 10.

Note that figure 10 in the original manuscript is now figure 9.

We disagree with the reviewer, differences in the Danube river are not trivial. On the other hand, we believe that the sentence does not correctly underline the differences we are observing. In the Danube river basin, there are modest irrigation activities that have seasonal effects on discharge under both historical and future climate. Also, there is a decrease of river discharge induced by climate change. Despite the negative effect of climate change on river discharge, the influence of irrigation is higher in the future than in the historical period. This is a good example of how climate change can increase irrigation, even if there is no explosive increase of irrigated areas. We propose to change and extend the sentence so is more clear, line 323:

*The second class* **is illustrated by the Danube River basin (Figure 9-a, as well as the Dnepr and Nelson river basins, see Table S1), and** *corresponds to river basins with modest irrigation activities* **(i.e. average irrigated fraction lower than 1% during the historical or future period)** *and a* **decrease of river discharge with** *climate change.*

25. Table 1: Add "land" or "regions" after Irrigated and Non-irrigated in the header. It is confusing with Irr and NoIrr. Also, instead of Irr-NoIrr, provide the values for NoIrr so that the readers can compare the magnitudes.

Thanks for the observation. We decided to delete the NoIrr information so the table was smaller and had less data. On the other hand, we do not think it is a good idea to delete the

Irr-NoIrr row and change it by the NoIrr values, because the Irr-NoIrr data allows to identify the influence of irrigation.

We propose to leave the Irr-NoIrr row, and add a NoIrr row for every variable.

26. Figs. 1 and 3. Please add outlines of the major watersheds in these figures. It might be helpful to combine these two figures into a single 4-panel plot with irrigated areas and irrigation side-by-side.

Thanks for the advice. We will change the Figs. 1 and 3 into a single 4-panel plot, noted as Fig. 1. Regarding the watersheds outlines, as already mentioned in Observation 12 for reviewer 1, we prefer to not include the outlines in the figures, to avoid too much information in the plots.

27. Figs. 4, 5, and onward. Please label each panel a, b, c, … (separating irrigated and non-irrigated) and refer the respective panel in the text. I found it challenging to match statements to the correct panels.

We propose to add an additional indication on the column so it is more understandable. See response 20 and 22 to reviewer 1.

28. Fig. 10: It would be helpful to combine the panels based on the three classes in section 3.5. For example, place panels b, c, d, and e first with a heading for heavy irrigation activities, followed by moderate irrigation, and low irrigation basins. It will be helpful to provide 3-4 examples for each category of basins to ensure the results are consistent based on the classification. Also, please specify what was the criteria/threshold for the three classes (Ln 312).

Note that figure 10 in the original manuscript is now figure 9.

Thanks for this observation. Regarding the figure, we change the plot order so it follows the three classes, first for heavy irrigation, second for modest irrigation and negative climate change impact, and third for modest irrigation and positive irrigation influence.

We use three variables to classify the river basins: the average irrigated fraction during a period (historical or future), the effect of climate change on discharge, and the irrigation influence on discharge. We explain every class:

1. Any basin with an average irrigated fraction higher than 1% during the historical or future period is considered as heavily irrigated. These are the main group, as we expect the most changes in these basins. We show results for the Nile, Rio Grande, Indus and Ganges, but there are others that we note in Table S1.
2. For the modestly irrigated basins (average irrigated fraction less than 1% for any period), we focus on those where climate change decreases the discharge. This group has fewer members, but is important because it shows the complex interactions between irrigation and a decrease of water resources. We present results for the Danube river basin, in Table S1 we also note the Dnepr and Nelson river basins

3. Finally, the third group is constituted by river basins that are modestly irrigated, and where the discharge increases due to activation of irrigation in the Irr simulation. This group is interesting as it shows the complex interaction between interaction and a changing climate in remote regions. We show results for the Senegal and Congo river basins, but we also note other basins in Table S1, for instance the Congo and the Chari river basins.

We propose to start each class by explaining briefly its main characteristics. We start by defining the variables used to classify each river basin, line 313:

*We classify changes in river discharge into three classes, **based on using three variables: the average irrigated fraction during a period (historical or future), the effect of climate change on discharge, and the irrigation influence on discharge.***

Also, line 313 of the original manuscript:

*The **first class** corresponds to large river basins with heavy irrigation activities, **i.e. with an average irrigated fraction higher than 1% during the historical or future period (illustrated by the** Nile, Rio Grande, Indus and Ganges; see **Figure 9-b, c, d, and e**).*

Also, line 322 of the original manuscript:

*The second class **is illustrated by the Danube River basin (Figure 9-e, as well as the Dnepr and Nelson river basins, see Table S1), and** corresponds to river basins with modest irrigation activities **(i.e. average irrigated fraction lower than 1% during the historical or future period)** and **a decrease of river discharge with** climate change.*

Finally, line 329 of the original manuscript:

*The third class **also** corresponds to river basins **with modest** irrigation activities **(irrigated fraction lower than 1%)** in both periods, **but** a slightly positive influence of irrigation on discharge.*

Regarding the Table S1, we changed the colors of certain basins that had been misclassified.

---

## Author Comment (AC2)

**RC2**: 'Comment on esd-2024-41', Anonymous Referee #2, 23 May 2025 reply

**Review of the manuscript 'Joint evolution of irrigation, the water cycle and water resources under a strong climate change scenario from 1950 to 2100 in the IPSL-CM6' by Arboleda-Obando et al, submitted to Earth System Dynamics**

**General summary**

The study of Arbeleda-Obando et al applies their recently developed irrigation scheme in a climate change study using an AMIP style simulation setup. They find that irrigation increases strongly in their future projection due to the increase in radiative forcing and land use. Furthermore, and additional to the impacts of climate change, irrigation is shown to enhance evaporation and thereby the hydrological cycle and impacts also regions outside of the irrigated areas.

This study is the logical next step after presenting the development and validation of their irrigation scheme in a uncoupled land surface simulation with fixed atmospheric forcing in their 2023 paper. It convincingly demonstrates interactions between the applied irrigation and the interactive atmosphere and highlights regions which are prone to increase in water stress and water reservoir depletion.

We thank the anonymous reviewers 2 for the time he spent reading and commenting on our paper. As for reviewer 1, we provide a point-by-point response to these comments. Sentences from the original submitted manuscript are presented in italic, while the proposition to respond to the observations are presented in bold. Lines correspond to the original manuscript.

**Detailed comments**

- 1. L105 If the energy balance is solved for the grid cell instead of the individual tiles, don't you reduce the effect of reduced surface temperature and higher surface humidity on the latent heat flux and thereby have a higher irrigation demand than you would have if you would compute the energy balance on tile? Do you think this effect would be significant?

We thank the reviewer for this observation. First, we must underline that we cannot elucidate the effect of computing the energy balance in the grid cell instead of individual tiles in the current IPSL-CM6, essentially because the atmospheric component only "sees" the grid cell but not the tiles in the lower atmosphere. This approach is called "simple flux aggregation" (de Vrese et al., 2016b). Note that the water balance is computed for every tile.

If the energy balance could be solved for individual tiles, lower surface temperature and higher surface humidity could induce a lower evaporative demand. If the result is a decrease of ET and an increase of soil moisture, ultimately irrigation demand could decrease, by a less intense irrigation rate or by a shorter irrigation season length. Besides, even if there is not decrease on irrigation demand (because the soil moisture is dry and water stress is high), changes in the evaporative demand could have an impact on the irrigation efficiency (i.e. the irrigation water fraction that is actually evaporated, see Jägermeyr et al., (2015)).

Then, we could observe an increase of runoff and recharge as return flows from the irrigated volume.

Note anyway that the "simple flux aggregation" approach in ORCHIDEE LSM partially represents the change in surface temperature and surface humidity, and is able to represent the effects on the water balance, but is not currently able to represent the sharp contrast in surface conditions for the energy balance. It is then difficult to speculate on the significance of the effect. In any case, we believe this shortcoming is part of the modelling uncertainties, and should be noted.

We propose to add a sixth position on the list that enumerates model shortcomings, line 406:

***6. The misrepresentation of subgrid variability in the land surface-atmosphere coupling (de Vrese et al., 2016b), specifically on the energy budget computed by IPSL-CM6, could have an impact on irrigation demand, on atmospheric feedbacks and on irrigation efficiency (de Vrese and Hagemann, 2018).***

And twe new references:

de Vrese, P. and Hagemann, S.: Uncertainties in modelling the climate impact of irrigation, Clim Dyn, 51, 2023–2038, https://doi.org/10.1007/s00382-017-3996-z, 2018.

de Vrese, P., Schulz, J.-P., and Hagemann, S.: On the Representation of Heterogeneity in Land-Surface–Atmosphere Coupling, Boundary Layer Meteorol, 160, 157–183, https://doi.org/10.1007/s10546-016-0133-1, 2016b

- 2. L109 You spend more time explaining the climate model than the irrigation scheme. Probably this information is available in your irrigation paper already, but please add how you avoid to accidentally irrigate too large parts of the grid cell. As you don't seem to use an individual tile for irrigated crops (line 94 mentions bare soil, forest and crops + grasses), the soil moisture deficit in a rather dry tile would constantly remain high because even if you add enough water to saturate the irrigated fraction of the tile, the mean tile state would still show a deficit in the next time step.

We thank the reviewer for this observation. The reviewer is right, irrigated crops don't have a specific soil tile. It means that the soil moisture deficit in a rather dry tile will remain high, and then the water demand will not decrease. Also, the reviewer is right that this information is available in the paper that presents the irrigation scheme. But we agree that listing the limitations that were already identified is important in order to discuss the results that we present hereafter. We propose to add a paragraph at the end of section 2.2, line 128:

***We note here some shortcomings identified in the irrigation scheme used here (Arboleda et al., 2024). The scheme represents a single irrigation technique (the flood technique), and uses a set of simplified rules to trigger irrigation and allocate available water. Besides, the scheme uses a joint representation of rainfed and irrigated crops within the same tile, and the scheme doesn't represent conveyance***

***losses. To restrain in part the effect of these shortcomings on estimated irrigation
volumes, parameter values were tuned by fitting the simulation to reported irrigation
datasets. But we must also note that this parameter tuning is overly simplistic, as it
uses globally uniform parameters. Despite these limitations the irrigation scheme
produces acceptable estimates of yearly estimation withdrawals at global scale, but
tends to underestimate irrigation withdrawals in China, India and the USA,
corresponding to the irrigation hotspots (Arboleda et al., 2024).***

- 3. L112 are the 0.65m root depth specific for crops or is this used everywhere?

This value is specific to crops, and corresponds to approximately 90% of the root system as
it is represented in ORCHIDEE, an is specific for crops. We add this information in line 112:

*Here, we briefly describe its main characteristics. First, the root zone depth is set according
to a user-defined parameter. In our case, we set this depth to 0.65 m (11 layers).* ***This depth
comprises approximately 90% of the crop root system as represented in ORCHIDEE.***

- 4. L116 you don't mention dams as water reservoirs for irrigation. Wouldn't it be
  important to include these, because they are used to mitigate the seasonality of river
  discharge and thus would enhance water availability during dry seasons?

Here we list the natural reservoirs that are represented in the model. But we agree that dams
are an artificial reservoir that could help to mitigate the seasonality of river discharge (stream
reservoir in ORCHIDEE) and thus enhance water availability. Besides, we think that the
irrigation scheme could benefit from an explicit representation of dams operation and human
water management. We propose then to include these ideas in the paper. Please see
observation 6 from Reviewer 1.

- 5. L294: I would either use `strengthening effect` or `positive feedback`, but `positive
  effect of climate change` sounds wrong if it is about needing more irrigation

We agree that this sentence is not clear. Here, we are referring to the trends that we observe
on time series of groundwater and stream reservoirs. We observe a different behaviour after
and before 2040. Before, there is a small decrease in water storage reservoirs in the Irr
simulation. After 2040, both simulations depict an increase in water storage, but the increase
is faster in the NoIrr simulation than in the Irr simulation. We propose to slightly change the
sentence, line 293.

*Note that the* ***effect*** *of irrigation* ***on water storage reservoirs*** *seems to counteract* ***the
increase induced by climate change in*** *irrigated areas before 2040, and* ***after 2040 the
increase of water storage (groundwater and stream reservoirs)*** *is slower in the Irr
simulation than in the NoIrr simulation.*

- 6. L324: Why the time lack? I would assume irrigation happens during the summer,
  but then precipitation increases during winter? Could this mean that you over-irrigate
  during summer (due to reasons explained above) which means you start the winter
  with a much wetter soil than in the non-irrigation simulations although the extra water
  should have been mainly transpired by the crops instead of being stored in the soil?

Yes, irrigation happens during summer in the northern hemisphere, and as noted below, there is probably an overirrigation, even if the irrigation scheme has some restrictions. We also agree that irrigation induces an increase of soil moisture and ultimately of precipitation in the Irr simulation compared to NoIrr.

One possible mechanism is that the additional soil moisture allows the crops to transpire more until the end of fall (remember that crops are similar to grasses in our simulation, so there is no harvest calendar). The additional water into the atmosphere could induce more precipitation during winter. It could indicate a limitation in the model (because there is no crop during fall and winter), or it could indicate a case where the crops calendar has two or more harvest seasons.

In any case, we agree that this particular case is especially interesting, but understanding the mechanism would need a focus on regional wind and precipitation patterns that goes beyond the scope of the manuscript. Besides, we consider that some of the uncertainties that arise from model limitations in this case are already mentioned (see the listed limitation from the modelling framework and from the irrigation scheme in section 4 of the original manuscript, also response to observation 1 from reviewer 2).

- 7. L421: You found precipitation increase due to irrigation mainly in regions very close to the irrigated regions. However, others studies, e.g. de Vrese et al. (2016) found rather strong remote effects for today's irrigation. Do you see these effects as well?

Note that figure 7 in the original manuscript is now figure 6.

We thank the reviewer for this observation. Our results suggest that some of these teleconnections are also present in our simulation, e.g. increase of P in central and West Africa, and central Asia (see Fig. 6-b, second column), and higher discharge in the Irr simulation in the Niger and Senegal rivers. But These changes are not easily detected, and we focused on impacts of irrigation on water resources rather than on a comprehensive analysis of teleconnections between irrigation and precipitation in far away regions.

This issue is important, and could need an effort to: to robustly define the areas that are prone to teleconnections between irrigation and precipitation, an assessment of the relationship between irrigation and atmospheric circulation, and the use of additional tools as back-trajectory methods and water tagging.

We propose to add a paragraph in line 422 to include these ideas (with two new references):

***Analyzing the atmospheric impact of irrigation in more remote regions is challenging and was not comprehensively addressed in this study. De Vrese et al. (2016a) tackled this issue by carefully studying how wind patterns are modified by irrigation in coupled simulations of historical climate, and showed that irrigation in India leads to increase P in East Africa and Central Asia, owing to water vapor advection and disturbances in the Asian Monsoon. Our results focused on the effects of irrigation on water resources, and therefore the effects on atmospheric circulation were not analyzed in detail, although similar processes may exist in our simulations. A better understanding of the effects of irrigation on precipitation in remote areas must define***

*what a "remote area" means, then analyze the relationship between irrigation (taking into account the intensity and spatial distribution of irrigation), and changes in atmospheric circulation and moisture transport in a given climate (either historical or future). This analysis can be more robustly complemented by the use of inverse trajectory methods (Wei et al. 2013), to estimate the contribution of irrigation to total precipitation, and by the use of water tagging, to track the origin of water vapor and continental recycling (Risi et al. 2013) and study the changes induced by irrigation.*

We also propose to add the reference in the introduction, line 40.

- 8. Fig 3: For consistency if would be nicer if you use gray either as zero change values in your colorbar or to indicate areas without irrigation and just use the continent outline as in the b) panel.

We appreciate this advice. But we think it is clearer to use gray to show the non-significant points according to the statistical test. That way, we can differentiate grid cells with small but significant values and grid cells where the change is not significant.

Using gray to indicate non-irrigated areas could also be confusing, since non-irrigated areas with non-significant changes would be represented in gray, while irrigated areas with non-significant changes would be represented in white. Non-significant areas would be identified by two different colors.

We therefore prefer to leave the use of gray as it is now, in order to be clearer in the information we present on the maps.

Finally, note that Fig 3 and Fig. 1 were composed into a new 4 panels plot, following observation 26 from reviewer 1.

- 9. Fig 4: You seem to have included the wrong maps: ET and R (which are also part of Fig 5) instead of precip and temperature as written in caption and text.

Note that figure 4 in the original manuscript is now figure 3.

We thank the reviewer for this typo. Figure 3 corresponds to precipitation and air temperature. The section correctly refers to P and Tas in section 3.3. We will correct the titles in the new version of the manuscript (reviewer 1 did the same observation, see observation 16 from reviewer 1).

- 10. Fig 6: why a fitted polynomial surface? Looks like a curve to me.

Note that figure 6 in the original manuscript is now figure 5.

The general method is a fitted polynomial surface. In this case, it produces a curve because we have a relationship  y = f(x) function, but the function Loess implemented in R can use up to 4 predictors.

We decided to cite the name of the method used to plot the curve.

- All Figs: all difference plots use exactly the same colorbar. While you can do so, of course, please at least mirror it depending on whether it depicts e.g. temperature is shown with colder in blue and warmer in red, which is intuitive, but for e.g. Precip an increase should be blue and not red, and vice versa

We will change color bars according to this advice.

**References**

- de Vrese, P., S. Hagemann, and M. Claussen (2016), Asian irrigation, African rain: Remote impacts of irrigation, Geophys. Res. Lett. 43, 3737–3745, doi:10.1002/2016GL068146.

---

## Author Response (AR2)

**Response to reviewer on the paper "Joint evolution of irrigation, the water cycle and water resources under a strong climate change scenario from 1950 to 2100 in the IPSL-CM6", in the Earth System Dynamics journal.**

Thank you for the revised manuscript. The expanded explanation throughout is appreciated, particularly the inclusion of model choices, simplifications, and limitations. The revision represents a significant improvement. I am recommending minor revisions, with a focus on the following two points:

We thank the anonymous reviewers for this second round to comment on our paper. Below, we provide a point-by-point response to these comments. Sentences from the manuscript submitted after the first review round are presented in italic, while the proposition to respond to the observations are presented in bold. Lines correspond to the manuscript indicated, submitted manuscript or manuscript with changes tracking.

1. (1) Quantitative interpretation of results: Many sections, particularly in the results (Sections 3.1 - 3.5), still rely heavily on qualitative language (e.g., "important increase," "slightly decrease," "no major change"). These descriptions can be subjective and would be significantly strengthened by including quantitative data (e.g., % changes, mm/year, or °C ranges). Please incorporate numerical values or trend magnitudes directly into the text to help readers better assess the significance of the findings without relying solely on visual interpretation of the figures.

Please read response to observation 6, 7, 8 and 10

2. (2) Language clarity and conciseness: There are multiple areas in the manuscript that could benefit from improved clarity and conciseness. Some phrasing is either redundant, awkward, or overly verbose. Few suggestions are noted below.

Please read response to observation 3, 4, 5 and 9

3. Throughout the manuscript, please remove 'see' before the Figure references.

We removed 'see' before the figure references, so it is more concise.

4. Line 87: Please remove: "We briefly describe each component." This is redundant and unnecessary, given the content that follows.

We removed the sentence "we briefly describe each component", so the manuscript is more concise

5. Line 109: Remove: "Organising Carbon and Hydrology In Dynamic Ecosystems". ORCHIDEE is already defined earlier in the manuscript.

We note that line 109 corresponds to the version with change tracking. In the submitted version, a second mention to the full ORCHIDEE acronyme is set in line 83. We remove the second mention of the acronyme, so it is more concise.

6. Line 132: "Remark that this simplification may have an impact on the timing and volume of total water withdrawal ..."

Please clarify/revise this statement.

We note that line 132 corresponds to the version with change tracking. In the submitted version, it corresponds to line 99.

Here, we wanted to note that the representation of crops as grassland has an impact on irrigation, because the irrigation module is not able to represent local crop calendars, and because harvesting is not included.

We propose to include these two points to clarify the sentence:

*"Remark that this simplification may have an impact on the timing and volume of total water withdrawal, by inducing an overestimation of water demand and ultimately water use.* **This is due to the lack of representation of the crop calendar, including the harvest stage, which keeps the leaf area index values high in the model.**

7. Line 209: The phrase "this scenario could be seen as the upper boundary of potential climate change impacts" is repeated thrice now. Please retain it at the first instance and expand on it in the Discussion, stating what is the implication of this choice for the model results.
Please consider adding a concise justification for the use of SSP5–8.5 in the first instance, e.g., it was chosen to represent a high-end pathway of climate forcing, allowing the analysis of irrigation under extreme water stress and strong climate signals, etc.

We use "upper boundary of potential climate change impacts" in lines 156, 449 and 526 of the submitted manuscript. We follow the reviewer recommendation, and add a sentence in line 156 to justify the use of the SSP5-8.5 scenario:

*The use of scenario SSP5-RCP8.5 could be seen as the upper boundary of potential climate change impacts and results in a strong global warming and important changes in precipitation.* **The use of this scenario allows for analysis of the interaction between irrigation and climate in a context of strong climate change signals and significant changes in irrigated land area***.*

And we add a sentence to explain the implications of this choice for model results, in line 449:

*The use of SSP5-8.5 as a single radiative forcing scenario induces a strong warming and significant changes in precipitation. This scenario could be seen as the upper boundary of potential climate change impacts, but the magnitude and spatial distribution of these changes are uncertain (AR6, IPCC, 2021).* **The direct impact of climate change on irrigation could be affected by this uncertainty, especially with regard to changes in precipitation patterns.**

8. Lines 212, 217, 220: The repeated mention of "Historical and SSP5–8.5" can be streamlined to improve readability. Consider rephrasing to avoid redundancy and improve flow.

We note that lines 212, 217 and 220 of the tracking changes document correspond to lines 156, 162 and 164 of the submitted manuscript.

We understand the concern of the reviewer, we propose to change the sentence in line 164 of the submitted manuscript, so readability is improved:

*Changes in land use include changes in cropland area (see Figs. S1 and S2 in the Supplementary Material). Each year of irrigated area per grid cell is also prescribed with LUHv2, using the same **setup (historical and SSP5-8.5 scenario)**.*

9. Line 280: 'processus' might be a typographical error.

Indeed, it corresponds to 'process'. This typo was already corrected.

10. Line 353 (and throughout Sections 3.1 - 3.5): Qualitative statements such as "Warming tends to be greater in northern latitudes" would be clearer if supported by specific values. For example, you can state that warming is more pronounced in northern latitudes, exceeding X°C in parts of Canada and Russia, compared to Y–Z°C in tropical regions.

We note that some of these qualitative statements are supported by quantitative results, shown in tables and figures. On the other hand, we understand that noting key values can further support the main ideas we are presenting. We propose then to add some key values in the manuscript, following the reviewer's observation.

In line 240 of the submitted manuscript (line 320 of manuscript with changes tracking):

*We observe that the influence of irrigation (i.e. the difference between the NoIrr and the Irr simulations) increases the average land values of ET, precipitation (P), runoff (R), and LAI **(+4%, +1%, +2% and +3%, respectively)**, while it depletes water storage in irrigated areas, i.e., groundwater storage (GWS**, -7%**) and stream storage (Stream S**, -8%**), but increases water storage in non-irrigated zones **(+2% for GWS)**. The influence of irrigation on total water storage (TWS) **on average land values** is positive **(+0.7%)**, which is partially due to an increase in soil moisture (SM**, +1% over land**).*

In line 258 of the submitted manuscript (line 346 of manuscript with changes tracking):

*In irrigated areas, precipitation may either increase due to climate change (e.g., China and southern India, **with local extremes of more than +1 mm/day**) or decrease (Mediterranean area, **with local extremes in the Iberian Peninsula below -1 mm/day**), whereas warming occurs in all areas. These changes in climate can contribute to changes in irrigation: positive changes in precipitation can increase available water and water resources while decreasing the soil moisture deficit and water demand. Negative changes in precipitation increase water demand, which could increase irrigation if water resources are available. Warming tends to increase water demand, but it should be noted that warming tends to be greater in northern*

*latitudes **(warming over 7 °C)** than in tropical and southern areas **(warming over 3 °C)** as a result of the land warming pattern, visible in non-irrigated areas (see Figure 3-b).*

Line 305 of the submitted manuscript (line 406 of manuscript with changes tracking):

*Figure 6 shows the spatial distribution of the influence of irrigation (Irr-NoIrr) in the future for ET and P. In the future period, irrigation always increases ET in irrigated areas **(with local extremes in the Indus river basin above +1 mm/d)**, and in many non-irrigated areas nearby (Figure 6-a, especially in central Asia and the African Sahelian band, **with values below +0.1 mm/d**).*

Line 312 of the submitted manuscript (line 416 of manuscript with changes tracking):

*Like for ET, irrigation mostly increases P, over both irrigated and non-irrigated areas, but on smaller surfaces (Figure 6-b, **with values below 0.5 mm/d**; the same figure including the oceans is shown in Fig. S3).*

Line 347 of the submitted manuscript (line 465 of manuscript with changes tracking):

*The effects are mostly negative for both variables in irrigated areas, but in the case of Stream S, depletion is more important in the grid cells containing large rivers **(with local extreme values under -50 mm in the Indus and Rio de la Plata rivers)** …*

Line 368 of the submitted manuscript (line 493 of manuscript with changes tracking):

*In these river basins, irrigation activities decrease discharge values throughout the year **(decrease in the future due to irrigation ranges between -4 up to -51% of discharge)** under both historical and future climate conditions …*

Line 378 of submitted manuscript (line 509 of manuscript with changes tracking):

*and a decrease of river discharge with climate change **(decrease due to climate change range from -1 up to -15% of discharge)**.*

And line 385 of submitted manuscript (line 519 of manuscript with changes tracking):

*in both periods, but a slightly positive influence of irrigation on discharge **(increase in the future due to irrigation ranges between +3 to +12%)**.*

11. In general, please consider revising phrases like "important differences," "slower increase," "no major change," "slightly higher", etc., by including corresponding numerical values (e.g., percentage increases in ET, mm/year changes in precipitation). This would allow a more objective assessment of the magnitude of change and make your interpretation more robust.

Following reviewer observations, we included numerical values so the assessment of the magnitude is more objective. Please see response to observation 10 above.

12. Line 522: Similarly, please quantify "slightly higher". For example: "...discharge values were slightly higher (by ~X%) in the Irr simulation compared to the NoIrr simulation under historical climate conditions."

Following the reviewer's observation, we included numerical values. Please see response to observation 10, the last correction corresponds to this part of the text.

13. Fig. 5 and 7: Different readers may interpret visual trends differently. Adding numerical summaries (e.g., trend magnitude) in the figure and the text would help standardize interpretation.

We included some numerical summaries in the text to standardize interpretation.

First, line 297 of the submitted manuscript (line 394 of manuscript with changes tracking):

*For ET, the NoIrr simulation shows a decreasing trend in irrigated areas **(from 1.81 mm/d during 1950-1975, to 1.79 mm/d during 1975-2000)** during the 1950-2025 period that is not present in the Irr simulation (Figure 5a, second row). Additionally, the changes in ET observed over land (Figure 5a, first row) are driven by changes in irrigated areas **(+0.15 mm/d in historical, +0.25 mm/d in future)**, as the ET values in non-irrigated areas are similar for both simulations (Figure 5a, third row). Finally, we observe that the increase in ET in irrigated areas after 2025 is faster in the Irr simulation than in the NoIrr simulation, even though irrigation expansion stops by 2040 (Figure 2-a). In the case of P, irrigation activities increase the yearly average values over land **around 0.04 mm/d** (Figure 5b, first row).*

And line 337 of the submitted manuscript (line 450 of manuscript with changes tracking):

*The impact of climate change induces a positive trend in water storage (Figure 7-a and b, first row), whereas irrigation decreases the average GWS and Stream S in irrigated areas in around **-14 and and -12% in the future, respectively** (Figure 7-a and b, second row) and slightly increases the GWS and Stream S in non-irrigated areas **around +2 and +0.02% in the future, respectively** (Figure 7-a and b, third row).*

14. Discussion: Thank you for stating both the sources of uncertainty and the limitations of the modeling framework. It will be helpful to briefly state the impacts of the uncertainties (604 - 625) on model findings similar to the simple explanations provided for ORCHIDEE limitations (Ln 629 647). Listing the uncertainties without offering context limits their usefulness to the reader.

We propose to add the following sentence in line 465 (line 626 of manuscript with changes tracking):

***These uncertainties quantitatively influence the entire simulated water cycle, so that our results are model-specific and must be confirmed using other land-atmosphere models and available observations.***

Finally, we changed the color bar of maps in the Supplementary, so colors are coherent with figures in the main manuscript. We also changed minor typos in the manuscript.

---

## Author Response (AR3)

**Technical changes on the paper "Joint evolution of irrigation, the water cycle and water resources under a strong climate change scenario from 1950 to 2100 in the IPSL-CM6", in the Earth System Dynamics journal.**

We thank the editor for acceptin the paper. We addressed the use of the word "signficant", so it refers to statistically significant. To do so:

- We changed significant by important in line 21 of the second round review
- We changed significant by important in line 34 of the second round review
- We changed significant by important in line 162 of the second round review

We also corrected two typos:

- We corrected a typo "significantly" by "significant" in line 320 of the second round review
- We corrected "run" by "ran" in line 570 of the second round review

Finally, we changed "Code" to "Code and data", and we added a reference to the dataset used in the study and available in a zenodo repository in line 574. The corresponding reference was also added to the references list.